# Allnighter pseudokinase-mediated feedback links proteostasis and sleep in *Drosophila*

**Shashank Shekhar** [1] ✉, **Andrew T. Moehlman**[1,7]**, Brenden Park**[2]**,
Michael Ewnetu**[1]**, Charles Tracy**[1]**, Iris Titos**[3]**, Krzysztof Pawłowski**[2,4]**,
Vincent S. Tagliabracci** [2,5] **& Helmut Krämer** [1,6] ✉

In nervous systems, retrograde signals are key for organizing circuit activity and maintaining neuronal homeostasis. We identify the conserved Allnighter (Aln) pseudokinase as a cell non-autonomous regulator of proteostasis responses necessary for normal sleep and structural plasticity of *Drosophila* photoreceptors. In *aln* mutants exposed to extended ambient light, proteostasis is dysregulated and photoreceptors develop striking, but reversible, dysmorphology. The *aln* gene is widely expressed in different neurons, but not photoreceptors. However, secreted Aln protein is retrogradely endocytosed by photoreceptors. Inhibition of photoreceptor synaptic release reduces Aln levels in lamina neurons, consistent with secreted Aln acting in a feedback loop. In addition, *aln* mutants exhibit reduced night time sleep, providing a molecular link between dysregulated proteostasis and sleep, two characteristics of ageing and neurodegenerative diseases.

Adaptation to an ever-changing world is a fundamental function of nervous systems. Depending on the scope of environmental changes, neuronal adaptation may affect just a few synapses[1,2] or alter transcriptional profiles of complex neuronal circuits[3]. An environmental change experienced by all organisms is the cycle between night and day. Disruptions of the circadian pattern of light input impose additional challenges on the *Drosophila* visual system[4-6]. In response to continuous light, photoreceptors adapt by downregulating the number of their synapses[4]. Adaptive responses in rhabdomere structures were revealed by mutations interfering with the Unfolded protein response (UPR) due to loss of BiP AMPylation[5] or by interference with endolysosomal trafficking[7]. The adaptive changes of the visual system are regulated by a feedback mechanism that receives input from the circadian system and relies on cellular stress pathways for maintaining homeostasis during its execution[4-7]. Conditions that dysregulate stress pathways may thus cause vulnerabilities during adaptive challenges, similar to observations of compromised proteostasis enhancing the

progression of neurodegenerative diseases[8-10]. Adaptation in the *Drosophila* visual system offers an accessible system for exploring the molecular mechanisms underlying such vulnerabilities.

Here, we investigated the role in structural plasticity and sleep of a previously uncharacterized Drosophila gene, *CG12038*, which encodes a member of the Divergent protein kinases (Dipk) family. Mammalian Dipk proteins (previously also known as Fam69A/B/C, Dia1/HASF, or Dia1R)[11-15] and their invertebrate homologs are characterized by a cysteine-rich kinase-like domain interrupted by a putative EF-hand and by their localization within the lumen of the secretory pathway[12,13]. We renamed *CG12038* to *allnighter* (*aln*) based on phenotypes we observe under conditions of constant light exposure, including the reversible loss of structural integrity of rhabdomeres and reduced postsynaptic responses. We find that *aln* encodes a functionally conserved, secreted pseudokinase that retrogradely regulates proteostasis in photoreceptors as part of a feedback loop that depends on photoreceptor activity. Furthermore, *aln* mutants display reduced

[1]Department of Neuroscience, UT Southwestern Medical Center, Dallas, TX; O'Donnell Brain Institute, Dallas, USA. [2]Department of Molecular Biology UT Southwestern Medical Center, Dallas, TX, USA. [3]Molecular Medicine Program, University of Utah, School of Medicine, Salt Lake City, UT, USA. [4]Department of Biochemistry and Microbiology, Institute of Biology, Warsaw University of Life Sciences, Warsaw 02-776, Poland. [5]Howard Hughes Medical Institute, Maryland, USA. [6]Department of Cell Biology, UT Southwestern Medical Center, Dallas, TX, USA. [7]Present address: Surgical Neurology Branch, National Institute of Neurological Disorders and Stroke, National Institutes of Health, Bethesda, MD, USA. ✉e-mail: Shashank.Shekhar@UTSouthwestern.edu; helmut.kramer@utsouthwestern.edu

total sleep and delayed sleep onset at night time, thus establishing a molecular link between proteostasis and the regulation of sleep, a connection increasingly appreciated in flies and mammals[16–19].

## Results

### Aln regulates photoreceptor structural plasticity

Rhabdomeres are a characteristic feature of *Drosophila* photoreceptors. Within some 60,000 tightly packed microvilli, rhabdomeres contain an estimated 90% of the photoreceptor plasma membrane[20] along with the proteins, including Rhodopsin (Rh1) and Transient receptor potential (Trp), that mediate early steps in the phototransduction cascade[21]. Extended light exposure triggers plasticity of photoreceptor synapses[4], but corresponding adaptive changes in rhabdomeres have not been assessed. We measured rhabdomere area after a continuous 3-day exposure to ambient light (LL). Compared to flies maintained under a 12 h light:12 h dark cycle (LD), rhabdomeres of flies exposed to LL shrank by about one third (Fig. 1a–d). Consistent with this size reduction, the levels of the rhabdomeric proteins Rh1 (Fig. 1e, f) and TRP (Fig. 1e, g) were also significantly decreased. Reduced rhabdomere size is a transient adaptive response, as sizes are recovered upon return to an LD

environment within 3 days (Fig. 1c, d). These data indicate that structural plasticity in photoreceptors is not restricted to synapses, but extends to rhabdomeres which house early steps in the phototransduction cascade.

Our previous work[5] indicated that tight regulation of proteostasis in the secretory system is critical for the transient remodeling of photoreceptors. The proposed localization to the ER lumen and $Ca^{2+}$-binding of Dipk proteins[12] suggested a possible role for these proteins in the regulation of proteostasis in the secretory system. To test for such a function of the Dipk-type Aln protein, we generated the $aln^1$ allele in which most of the region coding for the kinase-like domain is replaced by a DsRed cassette (see Methods). Flies homozygous for $aln^1$ were viable and lacked overt developmental defects.

Severe phenotypes emerged, however, upon exposure to constant light. After 3 days of LL, electron micrographs of $aln^1$ photoreceptors revealed disrupted rhabdomeres with large vacuoles, multivesicular bodies, and extracellular vesicles not observed in $w^{1118}$ controls (Fig. 1h-m, Supplementary Fig. 1). To quantify structural changes in the same cohort of flies over time, we used the deep pseudo pupil (DPP) as a measure of rhabdomere alignment and structural

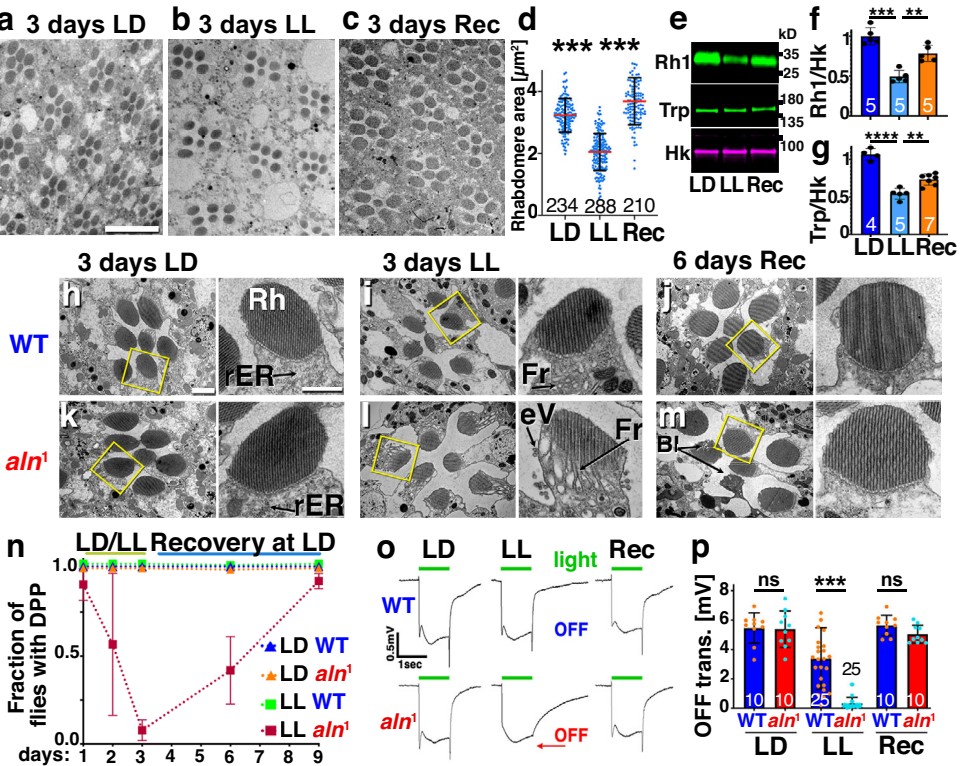

**Fig. 1 | Aln is required for photoreceptor structural plasticity. a–c** Rhabdomere sizes in TEM images of $w^{1118}$ flies after 3 days of LD (**a**) shrink after 3 days of LL (**b**) and recover when 3 days of LL are followed by three days of recovery (Rec) at LD (**c**). Scale bar in (**a**) is 20 µm and is the same for (**a–c**). **d** Scatter plots show means ± SD, numbers indicate n pooled from three independent flies. Significance threshold for *P*-Value are ***<0.001, as determined by one-way ANOVA with Bonferroni correction for multiple comparisons (*F* = 272). **e** Western blots of Rh1 Rhodopsin and TRP protein after 3 days of LL and recovery as quantified in (**f**; **g**); bar graphs show means ± SD and numbers of independent experiments. Significance threshold for *P*-Value are: **<0.01; ***<0.001 as determined by one-way ANOVA with Bonferroni correction for multiple comparisons. *F* values are **f**: 35.2; **g**: 54.5. **h–m** Representative TEM images from 3 separate flies each of $w^{1118}$ (**h–j**) or $aln^1$ (**k–m**) eyes after 3 days of LD (**h, k**), LL (**i, l**) or recovery (Rec) at LD (**j, m**) show the aberrant adaptation of $aln^1$ photoreceptors to LL and their recovery 6 days after return to LD. Scale bar in (**h**): 1 µm and is same for (**h–m**). Arrows point to rough ER

(rER), Fragmented rhabdomeres (Fr) or blemishes in recovered rhabdomeres (Bl). Yellow boxes indicate rhabdomeres shown in high magnification images. High magnification scale bars: 0.5 µm. (Supplementary Fig. 1 shows additional details of LL-treated *aln* photoreceptors). **n** Assessment of ommatidial structural integrity by deep pseudopupil (DPP) reveals the maladaptation of $aln^1$ flies to LL. *N* = 3 independent biological replicates with approximately 50 flies per genotype. Bar graphs show means ± SD. **o, p** Loss of OFF-transients in $aln^1$ flies is revealed by ERG traces of $w^{1118}$ and $aln^1$ females in response to a 1-s light pulse. Under LL, $aln^1$ mutants lose OFF transients (red arrow). The changes are reversed after 6 days of recovery (Rec). **p** Quantification of OFF transients as shown in panel (**o**), bar graphs show number of independent flies pooled from three replicates and means ± SD. *P*-values from two-way ANOVA followed by Bonferroni correction for multiple comparison test are (LD: 0.99; LL: 0.0007; Rec: 0.99; (*F* (2, 84) = 82). Genotypes are listed in Supplementary File 2. Source data are provided in Source Data file.

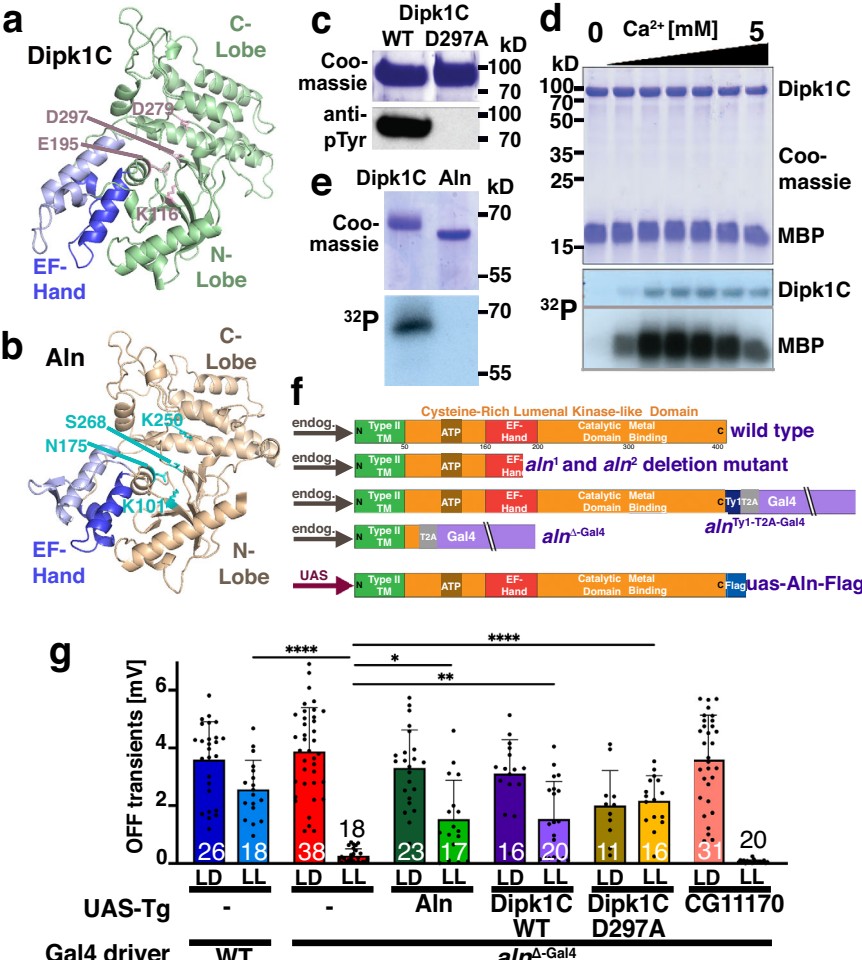

**Fig. 2 | Aln is a pseudokinase. a, b** Diagram depicting AlphaFold predicted kinase-like structures for human Dipk1C (**a**) and *Drosophila melanogaster* Aln (**b**). Four highly conserved residues in the kinase active site are indicated in the human Dipk1C, with only one (K[101]) being conserved in Aln. Structural superposition of Aln structure model against Protein Kinase A (PDB code: 1cdk) using the cealign algorithm yields Cα root mean square deviation (RMSD) of 5.3 Å over 152 residues. **c** Western blots using phospho-tyrosine antibodies visualized kinase activity for purified Dipk1C but not Dipk1C[D297A]. Equal loading was confirmed by Coomassie blue staining. **d** Kinase activity of Dipk1C is $Ca^{2+}$-dependent as visualized using $^{32}P$ phosphate incorporation into Dipk1C and myelin basic protein (MBP) in the presence of $Ca^{2+}$ concentrations ranging from 0 to 5 mM. **e** Unlike Dipk1C, purified wild-type Aln did not show kinase activity. Blots shown in c-e were repeated 3 times.

**f** Diagrams depict the genomic structures of *aln* alleles, including the predicted transmembrane domain, ATP binding site, and EF hand disrupting the conserved kinase-like domain. **g** Conserved function of Aln is revealed by the rescue of OFF-transients under LL condition by *aln*[Δ-Gal4]-dependent expression of Aln, Dipk1C and kinase-dead Dipk1C[D297A], but not CG11170. Bar graphs show number of flies pooled from three independent experiments and means ± SD. *P*-values in pairwise comparison to *aln*[Δ-Gal4] from Kruskal–Wallis test followed by Dunn's test for multiple comparisons are (wt: <0.0001; UAS-Aln: 0.024; UAS-DipkC1: 0.0041; UAS-Dipk1C[D297A]: <0.0001; UAS-CG11170: 0.41). ****$p < 0.0001$; ***$p < 0.001$; **$p < 0.01$; *$p < 0.05$. Genotypes are listed in Supplementary File 2. Source data are provided in Source Data file.

integrity in living flies[22]. Wild-type flies maintained their DPP under LL. By contrast, 93% (n = 150) of *aln1* flies lost rhabdomere integrity within three days (Fig. 1n). Upon return to LD, rhabdomere structure recovered with only few blemishes remaining (Fig. 1m). Quantification of DPP showed a nearly complete recovery in *aln1* flies that had lost rhabdomere structural integrity under LL (Fig. 1n).

We performed electroretinogram (ERG) recordings to further test the effect of loss of *aln* function on adaptation in the visual system. ERGs for LD-raised *aln1* flies were indistinguishable from wild type with respect to sustained negative potentials which reflect photoreceptor depolarization or OFF transients which reflect postsynaptic responses (Fig. 1o; Supplementary Fig. 2a). Under LL, however, OFF transients (Fig. 1o, red arrow) were almost absent (Fig. 1p). This effect was transient; OFF transients returned and were no longer distinguishable from wild type after a 6-day recovery in LD (Fig. 1o,p; Supplementary Fig. 2b). Dependence of this ERG phenotype on Aln function was

confirmed by functional rescue of the *aln1* mutant with a 4-kb genomic *aln* transgene which restored OFF transients under LL to near wild-type levels (Supplementary Fig. 2c).

Together these data indicate that Aln is required for functionally relevant homeostatic structural plasticity in photoreceptor neurons and that loss of Aln triggers a reversible maladaptation to extended light exposure. Despite the severe nature of these phenotypes, their reversibility distinguishes them from the progressive effects of neurodegeneration.

### Aln is a member of the DIPK family that lacks residues necessary for kinase activity

The AlphaFold Protein Structure Database[23] indicates a kinase-like fold for human Dipk1C and the *Drosophila* Aln ortholog (Fig. 2a,b), including the insertion of a EF-hand characteristic for the Dipk family[12]. Consistent with these structural features, purified human Dipk1C

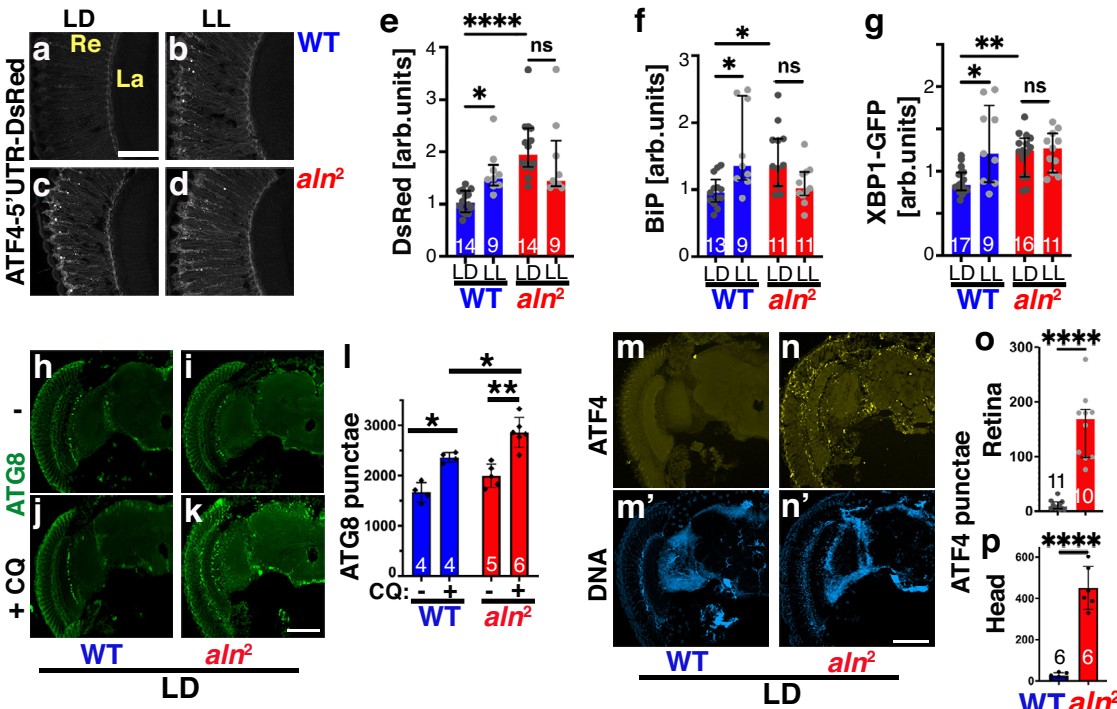

**Fig. 3 | Loss of Aln causes dysregulated proteostasis. a–d** Micrographs of cryosection of adult eyes of $w^{1118}$ (**a**, **b**) or $aln^2$ (**c**, **d**) flies treated for 3 days at LD or 1 day at LL and stained for the PERK activity reporter ATF4-5'UTR-DsRed. Quantified in (**e**), bar graphs show number of heads and median with interquartile range, Scale bar in a: 50 μm and is the same for (**a–d**). **f, g** Quantification of retinal staining for BiP (**f**) or the IRE1 activity reporter XBP1-GFP (**g**). Example images in Supplementary Fig. 4a–h. Flies were treated for 3 days at LD or 1 day at LL. Bar graphs show number of separate retinas pooled from three independent experiments and median with interquartile range. **e–g** P-values from Kruskal–Wallis test followed by Dunn's multiple comparison test are (**e**) WT,LD-LL: 0.014; $aln^2$, LD-LL:0.99; LD, WT-$aln^2$: <0.0001; (**f**) WT,LD-LL: 0.029; $aln^2$, LD-LL:0.14; LD, WT-$aln^2$: 0.0067; (**g**) WT,LD-LL: 0.023; $aln^2$, LD-LL:0.99; LD, WT-$aln^2$: 0.017. Significance threshold: $p < 0.05$; **$p < 0.01$, ****$p < 0.0001$. **h–k** $aln^2$ retinas and brains display elevated autophagic flux, $w^{1118}$ (**h**, **j**) or $aln^2$ (**i**, **k**) flies stained for ATG8. Flies were treated without (**h**, **i**) or

with CQ-containing food (**j**, **k**) to inhibit lysosomal degradation. Scale bar in (**k**) is 100 μm and is same for (**h–k**). **l** Bar graphs show quantification of Atg8 punctae in the $w^{1118}$ and $aln^2$ brains. Statistics: Kruskal–Wallis test followed by Dunn's multiple comparison test. Bar graphs show number of separate retinas pooled from three independent experiments and median with interquartile range, *$p < 0.05$; **$p < 0.01$. **m–p** Cryosection of adult heads of $w^{1118}$ (**m**) or $aln^2$ (**n**) flies stained for ATF4 (**m**, **n**) or DNA (**m'**, **n'**) as quantified in (**o**). Bar graph shows ATF4 levels in retinas (**o**) and heads (**p**) and the number of separate retinas pooled from three independent experiments. Statistical significance was assessed using 2-tailed $t$ test. Bar graphs show (**n**) and median with interquartile range. ****$p < 0.0001$. Scale bars in (**k**) and (**n'**) are 100 μm is same for (**m**, **n**). Re: retina, La: lamina. Scale bars in (**k**) and (**n'**) are 100 μm and is the same for (**h–k**) and (**m**, **n**). Genotypes are listed in Supplementary File 2. Source data are provided as a Source Data file.

protein displays kinase activity (Fig. 2c), that is regulated by $Ca^{2+}$ (Fig. 2d). In human Dipk1C, aspartate[297] ($D^{184}$ using protein kinase A nomenclature) is positioned as one of the signature residues conserved in active sites of kinases that binds a divalent cation to position the phosphates of ATP (Fig. 2a). Mutation of this residue to alanine abolished Dipk1C kinase activity (Fig. 2c). Interestingly, this as well as the additional active site residues $D^{279}$ (corresponding to $D^{166}$ in the prototypical PKA kinase domain) and $E^{195}$ ($E^{91}$ in PKA) are not conserved and replaced by $S^{268}$, $K^{250}$, and $N^{175}$ in Aln (Fig. 2b; Supplementary Fig. 3) which accordingly failed to show autophosphorylation (Fig. 2e). This suggests that Aln functions as a pseudokinase to regulate photoreceptor adaptation.

To further explore Aln function, we generated additional alleles (Fig. 2f). First, $aln^2$ is a null mutant lacking the 3xP3-DsRed cassette of $aln^1$ to facilitate imaging experiment in the visual system. Second, $aln^{Ty1-T2A-Gal4}$ is endogenously tagged with the Ty1 epitope followed by a self-cleaving T2A peptide and Gal4. OFF transients in ERG recordings from $aln^{Ty1-T2A-Gal4}$ under LL conditions are indistinguishable from wildtype controls (Supplementary Fig. 2d) indicating that the tagged Aln-Ty1 protein is functional. For functional rescue experiments, we generated the $aln^{\Delta-Gal4}$ null allele in which Gal4 instead of Aln is expressed from its endogenous promoter. Like $aln^1$, the $aln^2$ and $aln^{\Delta-Gal4}$ alleles display loss of OFF transients after 3 days of LL and recover when returned to LD conditions (Supplementary Fig. 2b, g). The OFF

transient phenotype of the $aln^{\Delta-Gal4}$ allele under LL was rescued when it was used to drive the expression of a UAS-Aln-Flag transgene (Fig. 2g).

To test functional conservation, we expressed human Dipk1C under control of the $aln^{\Delta-Gal4}$ driver and found that it restored OFF transients under LL (Fig. 2g). Importantly, expression of the kinase-inactive Dipk1C$^{D297A}$ mutant was also sufficient to restore OFF transients under LL (Fig. 2g). By contrast, expression of the *Drosophila* homolog of human Dipk2A/Dipk2B, CG11170, did not restore OFF transients under LL. Together, these findings indicate that Aln is a functionally conserved pseudokinase.

## Loss of *aln* dysregulates the UPR

The UPR is triggered by accumulation of misfolded proteins in the ER. In response, activated Ire1 promotes splicing of the transcription factor XBP1 as can be measured with a XBP1-GFP fusion[24] and activated PERK kinase phosphorylates eIF2 thereby reducing general translation but upregulating ATF4 translation as measured by a ATF4-5'UTR-DsRed fusion[25]. The combined activity of the activated XBP1, ATF4 and ATF6 transcription factors promotes the expression of a multitude of factors supporting protein folding that aid in restoring homeostasis. We previously showed that homeostatic UPR is a protective process necessary for photoreceptor adaptation[5]. LL environments trigger increased UPR in wild-type retinas as indicated by increased expression of the PERK activity reporter ATF4-5'UTR-DsRed (Fig. 3a, b, e), the ER chaperone

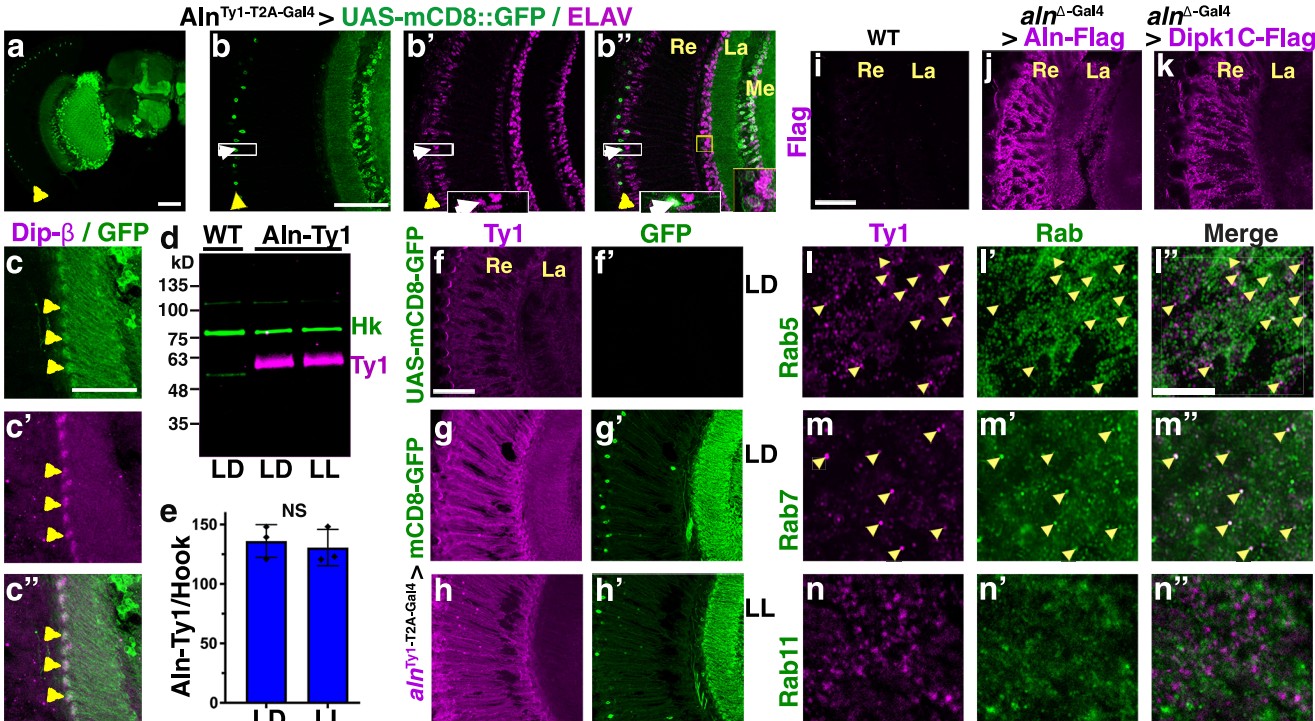

**Fig. 4 | Aln is retrogradely taken up by photoreceptors. a** Micrograph of adult hemi-brain expressing UAS-mCD8-GFP under control of $aln^{Ty1-T2A-Gal4}$ shows widespread expression of the *aln* gene in the brain and point to the layer of mechanosensory bristle neurons (arrowhead). Scale bar 40 μm. **b, c** Micrographs of adult heads expressing UAS-mCD8-GFP under control of $aln^{Ty1-T2A-Gal4}$ and stained for Elav (**b'**) or Dip-β (**c'**) show expression of the aln gene in many neurons, including retinal mechanosensory bristle neurons and L4 lamina neurons (arrowheads in **c**), but not photoreceptors. Enhanced example in white inset reveals dendrite and axon of mechanosensory bristle neuron. Scale bar in b is 40 μm, in (**c**) 20 μm. Experiments depicted in (**a**–**c**) were independently done 5 times. **d, e** Ty1-tagged Aln protein is detected by Western blot of lysates from adult heads of $aln^{Ty1-T2A-Gal4}$, LD, or LL treated as indicated and probed with antibodies against Ty1, $w^{1118}$ is used as negative control and Hook as loading control. **e** Quantification of blots for Ty1-tagged Aln relative to Hook protein ($n = 3$). Data are from three experimental repeats. Statistical significance using two-tailed *t* test. Bar graphs show mean ± SD. **f–h** Endogenous Ty1-tagged Aln is detected in lamina and retina in micrographs of LD (**f, g**) or LL (**h**) treated adult heads with UAS-mCD8-GFP and without (**f**) or with (**g, h**) $aln^{Ty1-T2A-Gal4}$ driver (**g, h**) stained for Ty1 and GFP. Scale bar in (**f**) is 40 μm for (**f**–**h**). **i–k** Exogenous Flag-tagged Aln (**j**) or human Dipk1C (**k**) expressed under control of the $aln^{Δ-Gal4}$ driver are detected in lamina and retina in micrographs of anti-Flag-stained adult heads, but not in wild-type control (**i**). Scale bar in (**i**) is 40 μm and is the same for (**i**–**k**). **l–n** In the retina, Ty1-tagged Aln partially colocalizes with Rab5-YFP (**l**), and Rab7 (**m**) but not Rab11-YFP (**n**). Scale bar in (**l**) is 5 μm for (**l**–**n**). Experiments depicted in (**f**–**h**) are repeated 3 times and l-n twice. Re: Retina, La: lamina, Me Medulla. Genotypes are listed in Supplementary Table 2. Source data are provided as a Source Data file.

BiP (Fig. 3f; Supplementary Fig. 4a, b) and the Ire1 activity reporter XBP1-GFP (Fig. 3g; Supplementary Fig. 4e, f). By contrast, $aln^2$ retinas displayed elevated levels of PERK activity (Fig. 3c, e), and BiP (Fig. 3f; Supplementary Fig. 4c) and IRE1 activity (Fig. 3g; Supplementary Fig. 4g) even under LD. Unlike wild type, however, $aln^2$ mutants failed to further upregulate these measures of the UPR response under LL (Fig. 3e–g; Supplementary Fig. 4a–h). Together these data indicate that photoreceptor adaptation to LL induces homeostatic UPR which requires *aln* function for its regulation.

### Elevated autophagy in *aln* mutants

The drastic reduction in rhabdomere size under LL (Fig. 1a–d) suggested a possible role of autophagy. Indeed, in LL both wild-type and $aln^2$ retinas showed elevated levels of autophagosomes identified as ATG8-positive punctae (Supplementary Fig. 4i–m), consistent with increased autophagy. To distinguish whether the elevated ATG8 levels under LL are due to blocked autophagic flux or increased induction of autophagy, we used chloroquine (CQ) to inhibit lysosomal degradation. In CQ-fed LL-treated flies, ATG8 levels were further increased in wildtype and $aln^2$ retinas (Supplementary Fig. 5a–e), arguing against blocked autophagic flux being the reason for elevated ATG8 staining. The canonical autophagy pathway was required for the LL-induced increase in the level of activated ATG8 (ATG8II), as RNAi-mediated knock down of ATG1, ATG5, ATG9, or ATG18 prevented the increase of ATG8II seen in wild-type flies (Supplementary Fig. 6).

Under LD, Rh1 and Trp proteins are largely confined to rhabdomeres that stretch along the cell bodies[21], giving rise to a "stringy" appearance in longitudinal sections (Insets in Supplementary Fig. 4n, p). Consistent with autophagy contributing to rhabdomere downsizing under LL (a process we call "rhabdophagy"), Rh1 (Supplementary Fig. 4o, q) and TRP (Supplementary Fig. 4s, u) were detected in discrete punctae that partially colocalized with the autophagosome marker ATG8 (Supplementary Fig. 4s, u). Rh1 protein levels decreased over time in wild-type and $aln^2$retinas exposed to LL (Supplementary Fig. 5f, g). At first glance, reduced Rh1 levels by western blot may appear contradictory to the elevated levels depicted in the immunofluorescence images (Supplementary Fig. 4o, q). It is important to note, however, that reduced antibody accessibility has been documented for the tightly packed proteins within rhabdomere membranes[26]. Therefore, loss of rhabdomere integrity concomitant with relocalization of rhabdomeric proteins to vacuoles seen by EM (Fig. 1i, l; Supplementary Fig. 1) is consistent with the observed elevated Rh1 immunofluorescence staining. Levels of ATG8 mRNA did not significantly change between LD and LL conditions (Supplementary Fig. 4v) pointing to post-transcriptional upregulation of autophagy under LL conditions.

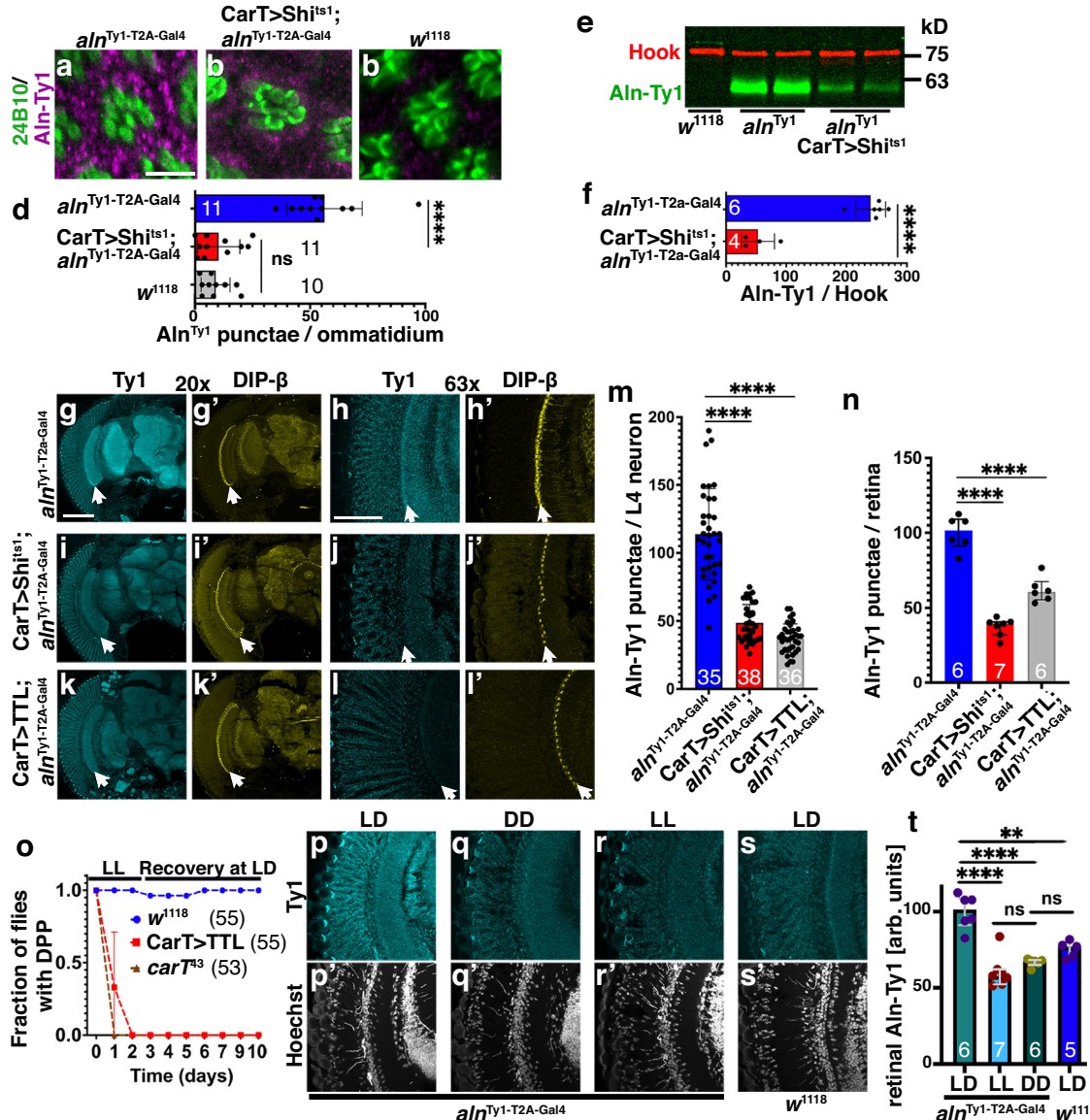

**Fig. 5 | Aln expression depends on photoreceptor synaptic activity.**
**a–c** Ommatidial cross-sections show Ty1-tagged Aln in photoreceptor cell bodies (**a**) reduced levels in photoreceptors expressing QUAS-Shi[ts1] (**b**) or $w^{1118}$ control (**c**) stained for chaoptin (24B10) and Ty1-Aln. Scale bar in (**a**) is 5 μm and is the same for (**a–c**). Numbers indicate n. **d** Quantification of Aln-Ty1 punctae per ommatidium. Bar graphs show mean ± SD, samples taken out of three experimental repeats. P-values for comparison to CarT>Shi[ts1]; $aln^{Ty1-T2A-Gal4}$ from one-way ANOVA with Bonferroni correction for multiple comparisons are: $w^{1118}$: 0.99; $aln^{Ty1-T2A-Gal4}$: <0.0001; $F(2, 29) = 58.24$. **e, f** Western blot of lysates from adult heads of $w^{1118}$, $aln^{Ty1-T2A-Gal4}$ ($n = 6$) without or with $CarT^{HA-T2A-QF2}$-driven QUAS-Shi[ts1] ($n = 4$) expression. Quantification of blots for Ty1-tagged Aln relative to Hook protein (**f**). Data are from three experimental repeats. Two-tailed t test yielded P-value < 0.0001. Bar graphs show number of independent samples and mean ± SD. **g–l** Compared to controls (**g**), Ty1-Aln levels are reduced in response to $CarT^{HA-T2A-QF2}$-drived retinal expression of QUAS-Shi[ts1] (**i, j**) or QUAS-TTL (**k, l**), especially in DIP-β marked L4 neurons (arrows in **g'–l'**). Quantification of Aln-Ty1 punctae in L4 neurons identified by DIP-β staining (**m**) or in the retina (**n**), bar graphs show number of L4 neurons (**m**) or retinas (**n**) pooled from three independent experiments and mean ± SD. P-values for comparison to $aln^{Ty1-T2A-Gal4}$ from one-way ANOVA with

Tukey's multiple comparisons test are (**m**) CarT>shi[ts1]; $aln^{Ty1-T2A-Gal4}$: <0.0001; CarT>TTL; $aln^{Ty1-T2A-Gal4}$: <0.0001; $F = 131$ (**n**) CarT>shi[ts1]; $aln^{Ty1-T2A-Gal4}$: <0.0001; CarT>TTL; $aln^{Ty1-T2A-Gal4}$: <0.0001; $F = 86.1$. **o** Compared to wild type, impaired photoreceptor synaptic activity in CarT$^{HA-T2A-QF2}$ > QUAS-TTL, and $CarT^{43}$ null flies causes reduced photoreceptor structural plasticity as LL triggers loss of deep pseudopupil (DPP). Graphs show number of flies pooled from two independent experiments and means ± SD. **p–s** Aln expression is sensitive to disruption of the circadian rhythm as indicated by Aln-Ty1 levels in adult heads (**p–r**) exposed for 3 days to LD, LL or DD as indicated, (**s**) wild type controls. **t** Quantification of Aln-Ty1 punctae in retinas as shown in (**p–s**), bar graphs show median with interquartile range and number of independent sections pooled from two experiments. For comparison to LD $aln^{Ty1-T2A-Gal4}$, P-values from one-way ANOVA with Bonferroni correction for multiple comparisons are: LL: 0.0001; DD: 0.0001; $w^{1118}$: 0.0012 ($F(3, 20) = 26.24$). Scale bar in (**g**) is 100 μm and the same for (**g, i, k**). Scale bar in (**h**) is 50 μm and the same for (**h, j, l, p–s**). Significance threshold for P-Values show in (**d–f, m, n, t**): ns, non-significant; *<0.05; **<0.01; ***<0.001; ****<0.0001. Genotypes are listed in Supplementary Table 2. Source data are provided as a Source Data file.

Induction of the *aln* phenotype by the switch from LD to LL suggests an involvement of the circadian system, in line with the disruption of the circadian clock observed under sustained high light intensities[27]. Therefore, we wondered whether the Aln LL phenotypes

may be mimicked in the $per^{01}$ circadian clock mutant. However, with regard to Rh1 levels and autophagosome induction, $per^{01}$ mutant under LD were indistinguishable from wild type or $aln^2$, with similar reduction in Rh1 and elevation of ATG8 punctae under LL (Supplementary Fig. 7).

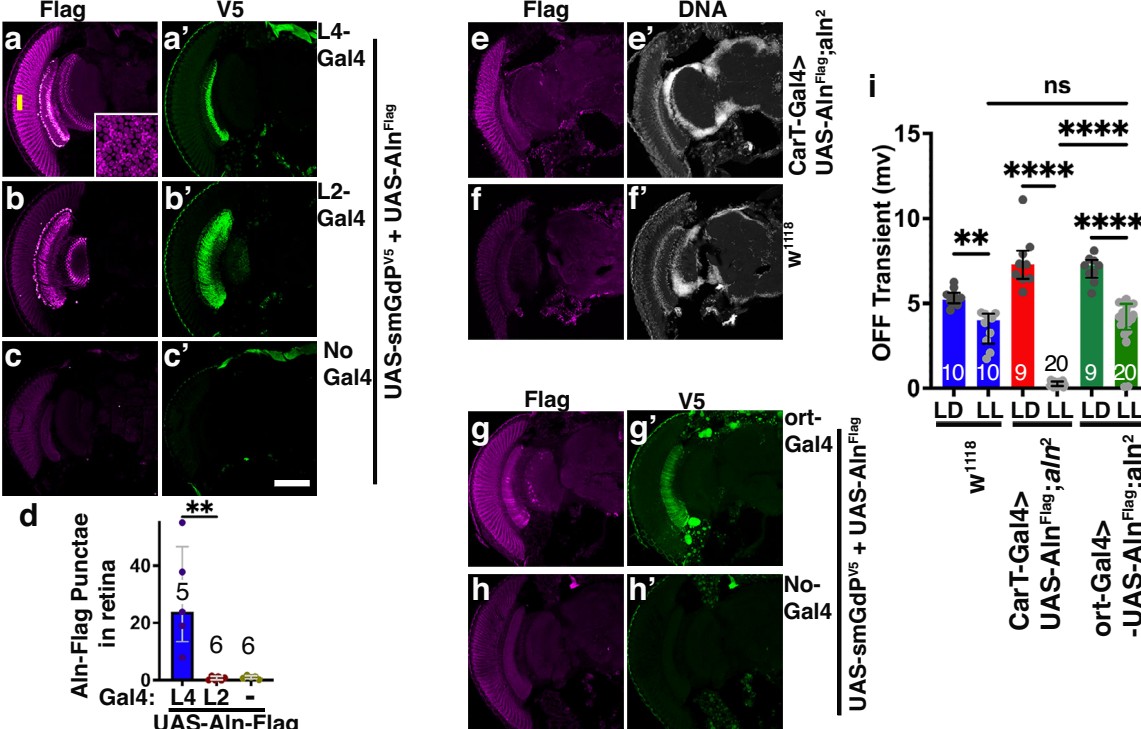

**Fig. 6 | Aln secretion from lamina neurons is necessary for photoreceptor plasticity. a–c** Aln-Flag is secreted from L4 lamina neurons but not L2 neurons when co-expressed with the cell-autonomous smGdP$^{V5}$ marker under control of L4-Gal4 (**a**) or L2-Gal4 (**b**) and undetected in the absence of a driver (**c**). **d** Quantification of Aln-Flag punctae in retinas of the indicated genotypes. For comparisons to L4-Gal4>uas-Flag-Aln, *P*-values determined by Kruskal–Wallis test followed by Dunn's test for multiple comparisons are for uas-Flag-Aln: 0.0023; for L2-Gal4>uas-Aln-Flag: 0.0014; *F* = 12.9. Bar graphs show median with interquartile range of number of punctate and number of separate sections pooled from two independent experimental repeats.; **p* < 0.01. **e, f** Aln-Flag is detected in photoreceptors when expressed under control of CarT$^{HA-T2A-Gal4}$ (**e**) but not in wild-type controls (**f**). **g, h** Aln-Flag co-expressed with the cell-autonomous smGdP$^{V5}$ marker is detected in photoreceptors and lamina neurons when expressed under control of

the lamina neuron specific ort-Gal4 driver (**g**) but not without driver (**h**). **i** Quantification of OFF transients indicated that expression of Aln-Flag in lamina neurons (Ort-Gal4) but not photoreceptors (CarT$^{HA-T2A-Gal4}$) rescues OFF transients under LL in *aln$^2$* mutants to the level observed in *w$^{1118}$* controls. Bar graphs show median with interquartile range and number of flies for each genotype and condition, pooled from three independent biological replicas. When compared to *w$^{1118}$*, LL, *P*-values from two-way ANOVA followed by Bonferroni correction for multiple comparison test are CarT-Gal4>uas-Aln-Flag; *aln$^2$*, LL: 0.0001; ort-Gal4>uas-Aln-Flag; *aln$^2$*, LL: 0.678; CarT-Gal4>uas-Aln-Flag; *aln$^2$*, LL: to ort-Gal4>uas-Aln-Flag; *aln$^2$*, LL: 0.0001; *F* = *F*(2, 70) = 14.27. Significance threshold for *P*-Values show in (**d**) and (**j**): ns, non-significant; *<0.05; **<0.01; ***<0.001; ****<0.0001. Scale bar in (**c'**) is 100 μm and the same for (**a–c, e, f**) and (**g, h**). Genotypes are listed in Supplementary Table 2. Source data are provided as a Source Data file.

---

Interestingly, proteostasis dysregulation in *aln$^2$* mutants was not restricted to photoreceptor cells. Elevated autophagic flux in many brain areas was evident in CQ-fed *aln$^2$* flies (Fig. 3h–l). Elevated levels of lipidated ATG8II were also detected in western blots of *aln$^2$* whole brain lysates (Supplementary Fig. 4w), suggesting brain-wide effects on autophagy rather than the effect being limited to the retina. Consistent with brain-wide proteostasis dysregulation, elevated ER stress responses in *aln$^2$* retina and brain were revealed by drastically increased number of punctae of ATF4, a downstream target of PERK activation (Fig. 3m–p).

## Aln is a secreted protein retrogradely taken up by photoreceptors

To identify the cells expressing *aln*, we used the *aln$^{Ty1-T2A-Gal4}$* allele (Fig. 2f) to drive expression of UAS-mCD8-GFP. The resulting GFP expression pattern revealed *aln* gene expression in many neurons of the lamina, medulla, other brain regions (Fig. 4a, b) and sensory neurons in appendages including wings and legs (Supplementary Fig. 8a, b). In the lamina, *aln$^{Ty1-T2A-Gal4}$* drove expression of UAS-mCD8-GFP (Fig. 4b) and *aln$^{Δ-Gal4}$* that of UAS-NLS-mCherry (Supplementary Fig. 8c) in neurons as indicated by the neuronal marker Elav. Expression driven by *aln$^{Ty1-T2A-Gal4}$* was elevated in a subset of cells with the morphology of L4 lamina neurons[28] which were further identified by the L4-specific marker DIP-β[29] (Fig. 4c). In the retina, however, *aln$^{Ty1-T2A-Gal4}$* driven

expression was confined to mechanosensory bristle neurons (identified by their unique shape and position in the retina;[30] (yellow arrowhead in Fig. 4a, b) but excluded from photoreceptors (Fig. 4b, g, h; Supplementary Fig. 8c), consistent with a previous genome-wide expression analysis of individual cell types in the *Drosophila* visual system[31].

The phenotypes observed in photoreceptors, in contrast to their lack of *aln* expression, raised the possibility that Aln functions as a secreted protein. Ty1-tagging of Aln was confirmed by western blot detection of Aln-Ty1 at the predicted size from adult head lysates (Fig. 4d); size or expression levels in whole head lysates were not noticeably changed under LL (Fig. 4e). In contrast to *aln$^{Ty1-T2A-Gal4}$*-driven UAS-mCD8-GFP expression, Ty1-tagged Aln protein was detected in retinal cells, under LD or LL conditions (Fig. 4f–h), and Aln protein localization included cell bodies of photoreceptor neurons (Fig. 5a). These data suggest that Aln expressed in the lamina is secreted and taken up by photoreceptor cells. To further test this notion, we used the *aln$^{Δ-Gal4}$* driver to express a UAS-Aln-Flag or UAS-Dipk1C-Flag transgenes. Like endogenously tagged Aln-Ty1, but unlike *aln$^{Ty1-T2A-Gal4}$*-driven UAS-mCD8-GFP (Fig. 4f'–h'), Flag-tagged Aln or Dipk1C proteins were secreted and taken up into the retina (Fig. 4i–k). Colocalization of Aln in the retina with early endosomes marked by Rab5 (Fig. 4l) and late endosomes marked with Rab7 (Fig. 4m), but not Rab11, a recycling endosome marker

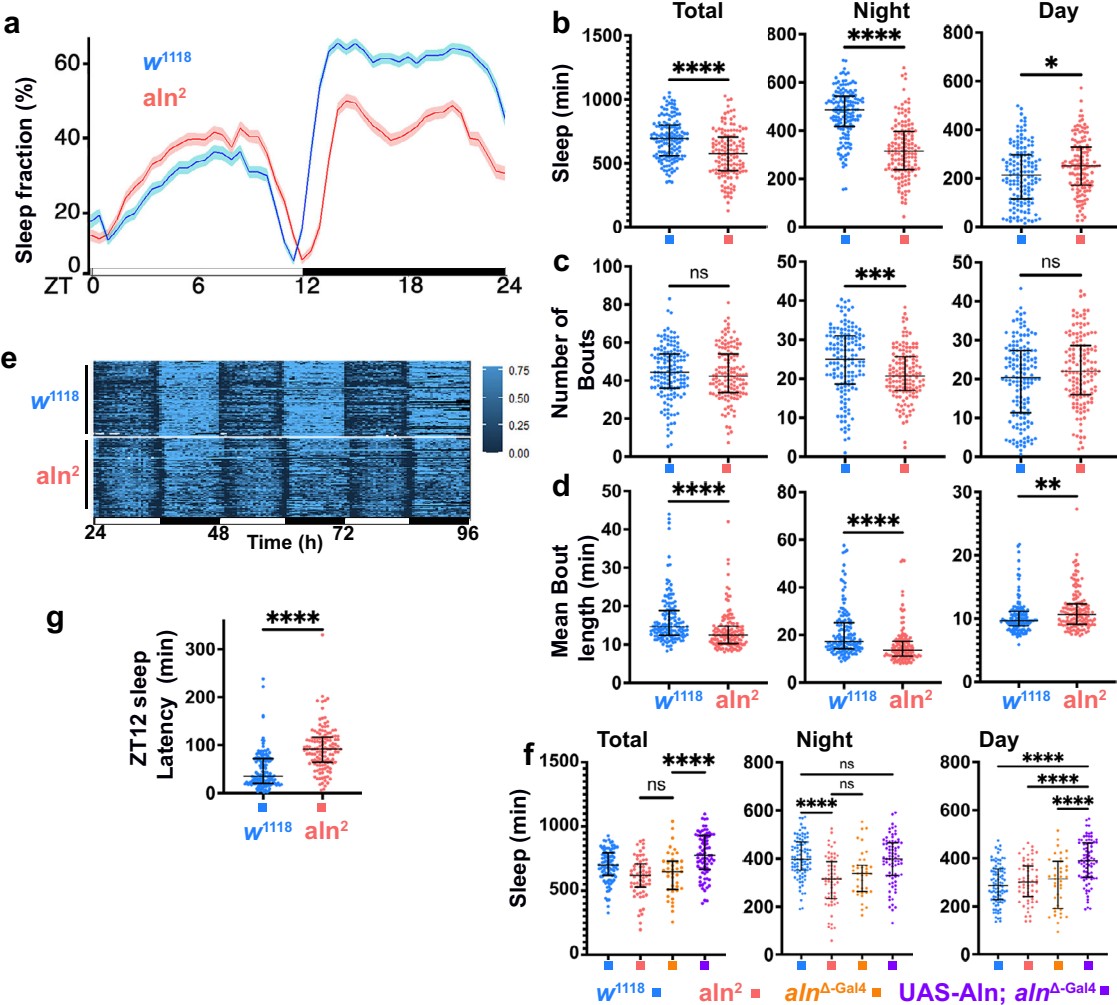

**Fig. 7 | Loss of Aln causes dysregulated sleep. a** Ethoscope recordings show reduced night and elevated day sleep of *aln²* compared to WT controls for a 24-h period under a 12 h light:12 h dark cycle LD pooled from 5 separate experiments with at least 20 flies each (a total of 147 WT controls and 137 *aln²* flies). From these recordings were measured total sleep (**b**), sleep bout number (**c**), and mean sleep bout duration (**d**) for the 24-h cycle (left), night (middle), and day (right). Scatter plots show median with interquartile range. Statistics for (**b**–**d**): Unpaired two-tailed *t* test. **e** Individual sleep patterns (at least 20 each) reveal their individual variability and the delay in both dawn and dusk activity of *aln²* flies compared to wild-type controls. Scale indicates 0 to 75% of time sleeping. **f** Sleep for *w^1118* (*n* = 91), *aln²*(n = 60), *aln^Δ-Gal4* (*n* = 39) and UAS-Aln; *aln^Δ-Gal4* (*n* = 79) flies shows no difference between the *aln²* and *aln^Δ-Gal4* alleles. Sleep of *aln^Δ-Gal4* is restored by UAS-Aln expression. Scatter plots indicate median with interquartile range for total sleep and sleep at night or day. Data are pooled from two independent experiments. Significance threshold for *P*-Values: *<0.05; **<0.01; ***<0.001; ****<0.0001 as determined by one-way ANOVA with Bonferroni correction for multiple comparisons; *F* values for total sleep (16.38); night sleep (15.83), day sleep (19.4). **g** Latency of sleep after ZT12 for *aln²* and wild-type flies. (*n* = 147 WT controls and 137 *aln²*). Graph show median with interquartile range. *P*-value is $p < 0.0001$ as determined by unpaired two-tailed *t* test. Genotypes are listed in Supplementary Table 2. The data used in this analysis are available in the Zenodo database under accession code (DOI: 10.5281/zenodo.7843827).

(Fig. 4n), was consistent with endocytosis and retrograde uptake of Aln into photoreceptor cells.

### Aln expression in L4 lamina neurons and brain depends on photoreceptor activity

To further explore the retrograde uptake of Aln, we blocked endocytosis into photoreceptors by expressing the dominant-negative Shi^ts1 protein using the photoreceptor-specific *CarT*^HA-T2A-QF2 driver (Supplementary Fig. 9d, e). Blocking endocytosis resulted in loss of Aln in photoreceptors (Fig. 5a–d). Unexpectedly, Shi^ts1 expression specifically in photoreceptor cells lowered Aln-Ty1 expression in whole-brain lysates as detected by western blot (Fig. 5e, f). This significant decrease in Aln-Ty1 levels suggested the effect may not be limited to the visual system. To test this, we analyzed Aln-Ty1 levels in brain sections of flies with photoreceptor-specific expression of Shi^ts1 (Fig. 5g–j). In addition

to reduced retinal localization, we observed decreased Aln-Ty1 staining throughout the brain, consistent with western blot results. The decrease was most striking in L4 lamina neurons (arrows in Fig. 5g–l). DIP-β counterstaining was used to specifically quantify Aln-Ty1 staining within L4 lamina neurons (Fig. 5m). Because Shi^ts1 affects synaptic recycling as well as general endocytosis, we wanted to separate these effects and used a QUAS-Tetanus toxin light chain (TTL) transgene[32] driven by CarT^HA-T2A-QF2 to inhibit synaptic vesicle fusion in photoreceptors. Silencing of photoreceptor synaptic activity was sufficient to suppress Aln-Ty1 expression in L4 lamina neurons (Fig. 5k–m) and photoreceptors (Fig. 5l, n) indicating that Aln expression is in part regulated by photoreceptor activity.

To further test the activity dependent uptake of Aln into photoreceptors, dark-raised flies were used and showed significant decrease of Aln signal in photoreceptors in comparison of LD maintained flies

(Fig. 5p–t), consistent with Aln uptake being activity dependent. Interestingly, constant light exposure also showed decreased Aln signal (Fig. 5r, t).

Together, these data suggest that the secreted Aln protein may be part of feedback loop in the Drosophila visual system; its expression is modulated by photoreceptor activity and, in turn, secreted Aln regulates the ability of photoreceptor neurons to adapt to long-term changes.

### Expression of Aln in lamina neurons rescues *aln* mutant phenotype

To further test the origin of Aln protein in the retina, we used Gal4 drivers specific to L4 and L2 lamina neurons to co-express UAS-Aln^Flag with UAS-smGdP^V5 as a cell-autonomous marker (Fig. 6a–c). Whereas UAS-smGdP^V5 faithfully marked L4 (Fig. 6a′) or L2 neurons (Fig. 6b), respectively, Flag-tagged Aln was detected in L4 and at high levels in the retina, when expressed in L4 neurons (Fig. 6a, d). By contrast, L2-Gal driven Aln-Flag expression was confined to L2 neurons (Fig. 6b, d), indicating a need for cell-type specific factors for Aln secretion.

To test the functional relevance of Aln secretion from lamina neurons, we compared functional rescue of ERG phenotypes by photoreceptor and lamina neuron-specific expression. CarT^HA-T2A-Gal4 driven expression of Aln-Flag (Fig. 6e, f) in photoreceptors did not restore OFF-transients of *aln*$^2$ mutants under LL (Fig. 6i). By contrast, ort-Gal4 which drives expression in lamina neurons (Fig. 6g′) yields Aln-Flag staining in lamina and photoreceptors (Fig. 6g, h) and restores OFF-transients in *aln*$^2$ mutants under LL (Fig. 6i).

Together these data demonstrate, that Aln is secreted from lamina neurons, especially L4, and endocytosed into photoreceptors as element of a feedback loop regulating photoreceptor plasticity.

Interestingly, blocking Aln uptake into photoreceptors by Shi^ts1 phenocopies the proteostasis defects observed in the retinas of *aln* mutants (Supplementary Fig. 9h–j) which further supports the functional importance of the Aln-mediated feedback loop. Furthermore, flies expressing TTL in photoreceptors as well as *CarT*[33] mutants, which both interfere with photoreceptor signaling to downstream lamina neurons, exhibit normal DPPs at LD. Upon transfer to LL, however, they lose their DPP, and fail to recover normal ommatidial structure when returned to LD (Fig. 5o). This adds further support to the importance of photoreceptor signaling as part of a feedback loop maintaining photoreceptor structural plasticity.

### Aln mutants have reduced sleep amount and delayed sleep onset

Because the circadian change in visual input contributes to the entrainment of the circadian clock and sleep, we investigated whether those behaviors were affected in *aln*$^2$ mutants using a video tracking system[34,35]. While the circadian periodicity of *aln*$^2$ mutants is unaltered compared to wild-type controls (Supplementary Fig. 10), the sleep patterns of *aln*$^2$ mutants were significantly altered. Data averaged from video tracking of flies in five separate experiments with at least 20 flies each (approximately 200 flies total) revealed significantly reduced total sleep with a reduction of night sleep and a slight increase in day sleep for *aln*$^2$ mutants (Fig. 7a, b). During night time, *aln*$^2$ mutants not only exhibited fewer sleep bouts (Fig. 7c) but the average duration of those bouts was significantly reduced compared to wild-type controls consistent with fragmented sleep and reduced sleep quality (Fig. 7d). Similar effects on sleep were observed in the *aln*$^{Δ-Gal4}$ null allele (Fig. 7f). Importantly, the reduced total and night time sleep caused by the *aln*$^{Δ-Gal4}$ allele was restored to control levels by expression of UAS-Aln under control of the aln promoter (Fig. 7f). Together these data indicate that loss of *aln* function causes reduced sleep and compromises sleep quality.

A second striking phenotype was evident from the display of individual sleep patterns, which revealed that the peaks of activity

periods typical for dawn and dusk were consistently delayed in *aln*$^2$ mutants (Fig. 7e). As a measure of this delay, sleep latency at nightfall was increased more than threefold for *aln*$^2$ mutants (Fig. 7g). Together, these data indicate that Aln function is not restricted to regulating adaptation in the visual system but is required for the regulation of sleep timing as well.

## Discussion

Photoreceptors exposed to extended periods of light display plasticity in rhabdomere size and engage neuronal stress responses, including UPR and autophagy. We find that loss of the secreted Aln pseudokinase compromises these responses and causes a reversible maladaptation. Aln is not required for development of the visual system or its responses under LD conditions. Only when photoreceptors are forced to adapt to extended light exposure is the vulnerability caused by loss of Aln function revealed. Under LL, active zones and rhabdomeres are downregulated (our data and[4,5,7]) while autophagy and the UPR are upregulated. In *aln* mutants, however, this upregulation is compromised resulting in similar rhabdomere dysmorphology as observed in mutations of *fic* or *BiP* that interfere with BiP AMPylation[5]. These mutations impair the tight control of the activity of the ER chaperone BiP[36–39], the central regulator of the UPR[40]. Loss of BiP AMPylation or Aln function thus exemplify mild disturbances in the proteostasis network that are well tolerated under stable conditions but are revealed as vulnerabilities during adaptive challenges, similar to observations with mutations in a subset of neuronal Rab proteins[7]. On a longer timescale, such vulnerabilities of the proteostasis systems are thought to contribute to challenges in neurodegenerative diseases and ageing[8,9,41].

The *aln* gene, despite being required for photoreceptor neurons to adapt to continuous light, is not expressed in photoreceptors but only in their target regions in the lamina and medulla in addition to more central brain regions. Aln may function in these brain regions to affect photoreceptor neurons indirectly. Alternatively, secreted Aln may directly affect photoreceptors, as we observe its uptake into early and late endosomes of photoreceptors. A non-autonomous function has also been suggested for Dipk2A, a human homolog of Aln previously known as DIA1, HASF or C3orf58, since a secreted form of Dipk2A has been observed to bind to insulin receptors and promote cardiomyocyte proliferation[11].

While the *aln* gene is not expressed in photoreceptors, their synaptic activity regulates Aln expression in downstream neurons, consistent with Aln being part of a feedback loop regulating structural plasticity in photoreceptors. Physiological significance of this feedback loop is supported by the rescue of *aln* phenotype from lamina specific expression in *aln* mutants. However, forced expression of *aln* in photoreceptor fails to rescue the phenotype, indicating that Aln is a lamina-derived signaling molecule that regulates homeostasis in photoreceptors, but the mechanism of such signaling remains to be investigated in future.

LL conditions also induce synaptic plasticity in photoreceptors by an *arrow*-dependent divergent canonical Wnt pathway which acts on the axonal cytoskeleton and downregulates the number of synaptic active zones[4,42]. Activation of this pathway depends on activity-dependent secretion of Wingless by glia cells[43]. This contrasts with Aln expression in lamina neurons, especially L4, but not glia (our data and[31]), suggesting that these are two separate feedback pathways regulating photoreceptor plasticity.

In addition to effects on visuals system adaptation, we find that Aln function is also required for the regulation of proteostasis networks in other brain regions and the amount and timing of sleep, with similar outcomes for *aln* null mutants. Changes in neuronal plasticity and proteostasis have been observed during sleep across organisms[19,33,44,45]. Timely degradation of key elements of the circadian clock are controlled by the ubiquitin proteosome system[46]

and chaperone-mediated autophagy[18]. Elevated autophagic flux during the sleep phase is thought to clear toxins and protein aggregates and support neuronal plasticity that contributes to memory consolidation[19,34,44,47–50]. Furthermore, forced changes in the level of autophagic flux alters the sleep-wake cycle of Drosophila[19]. Functional interactions have also been reported between sleep-wake cycle and the UPR, a third element of proteostasis networks. Sleep deprivation increases ER stress in mammals and invertebrates[51–53]. Altered expression of BiP can change sleep recovery[52] and PERK activity levels affect expression of the wake-promoting pigment dispersing factor (PDF)[17]. Despite all this evidence, it is still unclear, though, how cell-autonomous proteostasis pathways, like autophagy and the UPR, are coordinated across brain circuits to regulate the hallmarks of sleep, such as sleep amount and timing, reduced mobility and increased arousal threshold[54–56].

We provide evidence that Aln, a secreted protein, is involved in the regulation of sleep. Other secreted factors that play a role in addition to Aln, include neuropeptides[57,58], Nkt[59], and Nemuri[60]. Furthermore, the Drosophila genome encodes one additional Dipk-like protein and mammalian genomes possess a total of five members of the Dipk family[12].Given the function of Aln in the regulation of sleep, it will be interesting to further investigate whether any of these putative secreted enzymes are involved in sleep and proteostasis. Furthermore, it remains to be determined whether the effect of loss of Aln on sleep reflects its function in the visual system or its expression in other brain regions, including the mushroom body as seen in Fig. 4a.

The Dipk family of proteins is conserved from mammals to invertebrates. Its kinase-like fold was only belatedly recognized[12,15,61], in part due to the proposed $Ca^{2+}$-binding EF-hand within the N-lobe of some Dipk proteins (Fig. 2). Indeed, our analyses of human Dipk1C is consistent with both predictions: we observe kinase activity that is responsive to $Ca^{2+}$ levels and depends on $D^{297}$, a residue in the kinase motif conserved in mammals. Nevertheless, we find that this kinase activity is not required for the conserved function of Aln or human Dipk1C in facilitating photoreceptor adaptation. Indeed, two residues characteristic of active kinases, $D^{279}$ and $E^{195}$ in Dipk1C, are not conserved in Diptera. The biochemical mechanism by which these pseudokinases affect proteostatic processes in neurons remains to be elucidated, but our analyses highlight the non-autonomous nature of the signaling pathway regulated by Aln.

# Methods
## Biological reagents

| Reagent type (species) or resource | Designation | Source or reference | Identifiers | Additional information |
|---|---|---|---|---|
| Genetic reagent (D. melanogaster) | QUAS-Shi^ts | Bloomington Drosophila Stock Center | BDSC:30013; FBgn0003392; RRID:BDSC_30013 | |
| Genetic reagent (D. melanogaster) | QUAS-TTL | Bloomington Drosophila Stock Center | BDSC: 91808; FBgn0016753; RRID:BDSC_91808 | |
| Genetic reagent (D. melanogaster) | w^1118 | Bloomington Drosophila Stock Center | BDSC:3605; FBti0131930; RRID:BDSC_3605 | |
| Genetic reagent (D. melanogaster) | UAS-mCD8::GFP | Bloomington Drosophila Stock Center | BDSC: 32184; FBgn0003996; RRID:BDSC_32184 | |
| Genetic reagent (D. melanogaster) | UAS[Scer]-Xbp1-GFP.hg | Bloomington Drosophila Stock Center | BDSC:60731; FLYB:FBst0060731; RRID:BDSC_60731 | Sone et al. (2013) |
| Genetic reagent (D. melanogaster) | dsRed.crc(ATF4).5'UTR.tub | ref. 25 | FLYB: FBal0304834 | Gift from Don Ryoo, NYU. |
| Genetic reagent (D. melanogaster) | aln^1 | This paper | FBpp0072515 | |
| genetic reagent (D. melanogaster) | aln^2 | This paper | FBpp0072515 | |
| Genetic reagent (D. melanogaster) | aln^Δ-Gal4 | This paper | FBpp0072515 | |
| Genetic reagent (D. melanogaster) | aln^Tyl-T2A-Gal4 | This paper | FBpp0072515 | |
| Genetic reagent (D. melanogaster) | UAS-Aln-FLAG | This paper | FBpp0072515 | |
| Genetic reagent (D. melanogaster) | UAS-Dipk1C^WT-FLAG | This paper | NCBI Entrez Gene: 125704 | |
| genetic reagent (D. melanogaster) | UAS-Dipk1C^D297A-FLAG | This paper | NCBI Entrez Gene: 125704 | |
| Genetic reagent (D. melanogaster) | CarT^HA-T2A-Gal4 | This paper | FBgn0032879 | |
| Genetic reagent (D. melanogaster) | CarT^HA-T2A-QF2 | This paper | FBgn0032879 | |
| Genetic reagent (D. melanogaster) | Genomic Rab5-YFP | ref. 62 | | |
| Genetic reagent (D. melanogaster) | Genomic Rab11-YFP | ref. 62 | | |
| Genetic reagent (D. melanogaster) | GMR-dsRNA[white] | ref. 5 | DOI: 10.7554/eLife.38752 | P[GMR_wRNAi] |
| Genetic reagent (D. melanogaster) | UAS-Atg1-RNAi | Bloomington Drosophila Stock Center | BDSC: 26731 RRID:BDSC_26731 | |
| Genetic reagent (D. melanogaster) | UAS-Atg1-RNAi | Vienna Drosophila Research Center | VDRC ID: 16133 | |
| Genetic reagent (D. melanogaster) | UAS-Atg5 RNAi | Bloomington Drosophila Stock Center | BDSC: 27551 RRID:BDSC_27551 | |
| genetic reagent (D. melanogaster) | UAS-Atg5 RNAi | Bloomington Drosophila Stock Center | BDSC: 34899 RRID:BDSC_34899 | |
| Genetic reagent (D. melanogaster) | UAS-Atg18 RNAi | Bloomington Drosophila Stock Center | BDSC: 34714 RRID:BDSC_34714 | |
| Genetic reagent (D. melanogaster) | L4-Gal4 | Bloomington Drosophila Stock Center | BDSC: 49883 RRID:BDSC_49883 | |
| Genetic reagent (D. melanogaster) | Ort-Gal4 | Bloomington Drosophila Stock Center | BDSC: 56515 RRID:BDSC_56515 | = |
| Genetic reagent (D. melanogaster) | Ort-Gal4 | Bloomington Drosophila Stock Center | BDSC: 56517 RRID:BDSC_56517 | = |
| Genetic reagent (D. melanogaster) | L2-Gal4 | Bloomington Drosophila Stock Center | BDSC: 39912 RRID:BDSC_39912 | |
| Genetic reagent (D. melanogaster) | P[y + w + =5xUAS-DenMark::smGdP-V5]su(Hw)attP5 | Bloomington Drosophila Stock Center | BDSC: 62138 RRID:BDSC_62138 | |
| Antibody | anti-Hsc70-3 (BiP) (Guinea Pig polyclonal) | ref. 63 | FLYB: FBgn0001218; RRID: AB_2569409 | Gift from Don Ryoo, NYU (1:1000 IHC) |
| Antibody | anti-RFP (Rabbit polyclonal) | Rockland | Rockland:600-401-379; RRID:AB_2209751 | (1:500 IHC, WB 1:500) |
| Antibody | anti-GFP (Chicken polyclonal) | ThermoFisher Scientific | ThermoFisher Scientific:A10262; RRID: AB_2534023 | (1:1000 IHC) |
| Antibody | anti-GFP (Mouse monoclonal, B2) | Santa Cruz | Santa Cruz Anti-GFP Antibody (B-2): sc-9996 | (1:50 IHC, 1:250 WB) |
| Antibody | M2 anti-Flag (mouse monoclonal) | Sigma | Sigma:F-3165; RRID:AB_259529 | (1:1000 IHC, 1:2000 WB) |
| Antibody | Anti-GABARAP | Abcam | Abcam [EPR4805] (ab109364) | (1:200 IHC, 1:1000 WB) |
| Antibody | anti-Actin (mouse monoclonal) | Developmental Studies Hybridoma Bank | DSHB:JLA20; RRID: AB_528068 | 1:2000 (WB) |
| Antibody | Anti-HA (Mouse monoclonal) | MBL | mbl TANA2 HA antibody | (1:1000 IHC, 1:3000 WB) |
| Antibody | Alexa 488- or 568- or 647 secondaries Alexa Fluor 568 Goat anti-mouse Alexa Fluor 488 Goat anti-mouse Alexa Fluor 647 | ThermoFisher | Thermofisher fluorescent-secondary-antibodies | |

| | | | | |
|---|---|---|---|---|
| | Goat anti-mouse Alexa Fluor 488Goat anti-rabbit Alexa Fluor 568 Goat anti-rabbit Alexa Fluor 647 Goat anti-rabbit Alexa Fluor 647 Goat anti-guinea pig Alexa Fluor 568 Goat anti-guinea pig Alexa Fluor 488 Goat anti-guinea pig Alexa Fluor 668 Goat anti-rat Alexa Fluor 488 Goat anti-chicken | | | |
| Antibody | Goat anti-Mouse STAR-Red Secondaries | Abberior | https://www.fishersci.com/shop/products/star-red-goat-anti-mouse-igg/NC1933868 | |
| Antibody | IRDye 800CW Goat anti mouse IRDye 700DX Goat anti mouse IRDye 800CW Goat anti Rabbit IRDye 700DX Goat anti Rabbit | LICOR Biosciences | https://www.licor.com/bio/reagents/new-reagent=category=irdye-secondary-antibodies | |
| Antibody | STAR-Orange Secondaries | Abberior | | (1:500 IHC) |
| Antibody | STAR-Red Secondaries | Abberior | | (1:500 IHC) |
| Antibody | LICOR 800 or 700-secondaries | LICOR Biosciences | | (1:20,000 WB) |
| Antibody | Anti-Ty1 (mouse monoclonal, BB2) | Invitrogen | Thermo Fisher Scientific; MA5-23513 | (1:500 IHC, 1:2000 WB) |
| Antibody | Anti-RH1 (mouse monoclonal DSHB) | Developmental Studies Hybridoma Bank | https://dshb.biology.uiowa.edu/4C5 | (1:500 IHC, 1:1000 WB) |
| Antibody | Anti-24b10 (rabbit polyclonal) | Developmental Studies Hybridoma Bank | https://dshb.biology.uiowa.edu/24B10 | |
| Antibody | Anti-TRP (mouse monoclonal) | Developmental Studies Hybridoma Bank | https://dshb.biology.uiowa.edu/MAb83F6 | 1:300 IHC |
| Antibody | Anti-DIP-Beta (Guinea Pig polyclonal) | ref. 29 | Gift from Dr. Matthew Pecot, Harvard Medical School | 1:250 IHC |
| Antibody | Anti-ATF4 (rat polyclonal) | ref. 64 | Gift from Dr. Joseph Bateman, King's College | 1:2000 IHC |
| Antibody | Anti-ELAV (mouse monoclonal) | Developmental Studies Hybridoma Bank | https://dshb.biology.uiowa.edu/Elav-9F8A9 | 1:10000 IHC |
| Antibody | Anti-Hook (rabbit polyclonal) | | Lab generated | 1:5000 WB, 1:1000 IHC |

## Fly stocks and genetics

Bloomington Stock Center provided the following stocks: $w^{1118}$ (BS# 3605), UAS-mCD8::GFP (BS#32184), HS-Cre/Sco(BS#11092), QUAS-Shi[ts] (BS# 30013), QUAS-TTL(BS# 91808), L2-Gal4 (BS# 54949), L4-Gal4 (BS# 49883), Ort-Gal4 (BS# 56515 and 56517) and UAS-smGdP-V5 (BS# 62138) stocks. QUAS-Shi[ts] and QUAS-TTL were recombined with $CarT^{HA-T2A-QF2}$, and UAS-smGdP-V5 with UAS-Aln-Flag, and UAS-mCD8::GFP with $aln^{Ty1-T2A-Gal4}$. QUAS-Shi[ts] and QUAS-TTL after recombining with $CarT^{HA-T2A-QF2}$ (QUAS-Shi[ts], $CarT^{HA-T2A-QF2}$) were crossed with $aln^{Ty1-T2A-Gal4}$ to make stable stocks. The CRISPR/Cas9-engineered $aln$ alleles $aln^1$, $aln^2$, $aln^{Ty1-T2A-Gal4}$, and $aln^{\Delta-Gal4}$, the transgenes UAS-Aln-Flag, UAS-Dipk1C$^{WT}$-Flag, and UAS-Dipk1C$^{D>A}$-Flag and the photoreceptor-specific drivers CarT$^{HA-T2A-Gal4}$, and $CarT^{HA-T2A-QF2}$ were generated in this study. RNAi lines are described in Key resources table and were obtained from Bloomington Stock Center and the Vienna Drosophila Resource Center. The ATF4$^{5'UTR}$-dsRed[25] and the Xbp1-GFP[24] lines were a gift from Dr. Don Ryoo (NYU) and were crossed into the desired genetic backgrounds. Experiments employing CarT$^{HA-T2A-QF2}$ driven QUAS-Shi[tsl] expression to inhibit endocytosis were conducted at 25 °C, a temperature effective in photoreceptors as confirmed by ERGs and

consistent with previous results showing that Shi[tsl] expression can change photoreceptor morphology as low as 19 °C[65].

## Generation of Aln transgenes

For the genomic rescue transgene, a 3.6-Kb DNA fragment containing 1 kb upstream of Aln was amplified by PCR (for primers see: Supplementary file 1) which added KpnI and BamHI sites for cloning into a pCasp-AttB transformation vector confirmed by sequencing and inserted into the 59D3 AttP landing site[66]. Flag-tagged UAS-Aln and human Dipk1C transgenes were inserted into a pUASt-AttB vector, confirmed by sequencing and inserted into 43A1 AttP landing site[66]. Transgenic flies were generated by BestGene, Inc.

## CRSIPR/Cas9 generated Aln alleles and CarT drivers

The endogenously tagged $aln^{Ty1-T2A-Gal4}$ gene, the $aln^1$, $aln^{\Delta-Gal4}$ mutants and the $CarT^{HA-T2A-QF2}$ and $CarT^{HA-T2A-Gal4}$ drivers were generated essentially as described[67] using CRISPR/Cas9 tools available from the O'Connor-Giles, Wildonger, and Harrison laboratories[68]. Specifically, gRNAs (see Supplementary File 1) were introduced into the pU6-BbsI vector and co-injected with the appropriate template plasmid for homologous repair. Embryo injections were done by Bestgene (Chino Hills, CA), and the resulting, potentially chimeric adult flies were crossed with $w^{1118}$ flies. The vas-Cas9(X) line (BS55821) was used for CRISPR/Cas9 injections.

For the $aln^1$ null allele, the template plasmid was assembled in the pHD-DsRed backbone using approximately 1 kb PCR-amplified 5' and 3' homology arms. Potential founders were crossed to $w^{1118}$ and resulting flies with eye-specific DsRed expression[69] were selected, confirmed by PCR, balanced and homozygotes collected for further analysis. For the generation of the $aln^2$ allele, the DsRed cassette was removed from $aln^1$ using Cre expression, resulting lines confirmed using PCR, balanced and homozygotes collected for further analysis.

For the $aln^{\Delta-Gal4}$ and $aln^{Ty1-T2A-Gal4}$ alleles, the template plasmids were assembled in the pBS-KS backbone using approximately 1 kb PCR-amplified 5' and 3' homology arms. Potential founders were crossed to UAS-mCherry (BS: 52269) and resulting flies with $aln^{Ty1-T2A-Gal4}$-driven mCherry were selected, confirmed by PCR and balanced.

For the $CarT^{HA-T2A-QF2}$ and $CarT^{HA-T2A-Gal4}$ drivers, the template plasmids encoding a 3xHA tag followed by a T2A self-cleaving peptide and QF2 or Gal4, respectively, were assembled in the pBS-KS backbone using approximately 1 kb PCR-amplified 5' and 3' homology arms. Potential founders were crossed to QUAS-mCherry (BS: 52270) or UAS-mCherry (BS: 52269) reporters and the resulting flies with retinal mCherry expression were selected, balanced and in-frame fusions of the 3xHA-T2A-Gal4/QF2 tags with CarT were confirmed by sequencing PCR products.

See Supplementary File 1 for sequences of gRNAs and confirmation primers.

## Fly rearing conditions

All flies used for the experiments were reared on standard molasses fly food, under room temperature conditions. For light treatments, ambient light intensity of 700 lux was used at 25 °C. All flies were aged 3-5 days, and placed in 5 cm diameter vials containing normal food, with no more than 25 flies, and placed at either LD (lights ON 8am/lights OFF 8 pm) or LL. ERGs and head dissections were performed between 1 PM and 4 PM. For the Shi[ts]-mediated photoreceptor silencing experiments, all experiments were done at 30 °C.

## Electron Microscopy

Flies were reared exposed to 3 days of LL or LD as described above. Half of the LL cohort were returned to the LD cycle for 6 days for recovery. Fly heads were dissected in ice-cold fixative containing 2% paraformaldehyde, 2% glutaraldehyde in 0.1 M phosphate buffer to remove the proboscis. An incision in the posterior cuticle facilitated

access to fixative in which flies remained for 24 hrs before further processing. Fixed heads were thoroughly washed in 0.1 M phosphate buffer and 0.1 M sodium cacodylate buffer before embedding in 3% agarose. Embedded heads were then post-fixed for 1.5 hrs in 1% osmium tetroxide and 0.8% potassium ferricyanide in 0.1 M sodium cacodylate buffer at room temperature, washed with water three times, then stained with 4% uranyl acetate in 50% ethanol for 2hrs. Samples were dehydrated by increasing ethanol concentrations then transitioned into Embed-812 resin starting with propylene oxide and increasing resin concentrations in propylene oxide. Freshly prepared resin was used to embed heads before baking at 60 °C overnight. Thick ( ~ 0.5 μm) sections were cut, stained with toluidine blue and imaged to confirm appropriate section depth. Thin sections (70 nm) were then cut using a diamond knife (Diatome) on a Leica Ultracut 6 ultramicrotome. Two thin sections were placed on a formvar-coated copper grid before a final staining with 2% uranyl acetate followed by lead citrate. Images were procured using a JEOL 1400 Plus (NIH grant 1S10OD021685-01A1) transmission electron microscope.

## Rhabdomere size quantification

Transmission electron micrographs taken from $w^{1118}$ heads at 250X zoom were opened in ImageJ (NIH), scaling was properly set, and threshold was adjusted to highlight rhabdomeres in ommatidia with trapezoidal shape at roughly 90-100 μm depth which in these sections corresponded to the 4th and 5th rows from the lens. Rhabdomeres were quantified using the particle analyzer to measure the area of individual rhabdomeres from thresholded images. Only R1-R6 data were used in the dataset, as R7s were significantly smaller. At least 36 rhabdomeres were quantified from three separate flies for each light treatment cohort. Data were imported into Prism 8 for statistical analysis.

## Electroretinograms

ERGs recordings were done as described previously[21]. In brief, glass electrodes filled with 2 M NaCl were placed in the fly thorax (reference electrode) and surface of the corneal lens (recording electrode). A computer-controlled LED light source (MC1500; Schott, Mainz, Germany) was pulsed for 1 s at 4 s intervals. The traces of ERG were collected by an electrometer (IE-210; Warner Instruments, Hamden, CT), digitized with a Digidata 1440 A and MiniDigi 1B system (Molecular Devices, San Jose, CA), and recorded using Clampex 10.2 (Molecular Devices) and quantified with Clampfit software (Molecular Devices). Recordings were done in batches of ten to twelve and resulting quantification for each genotype are pooled from three independent and blinded biological replicas.

## Deep pseudopupil analysis

$CO_2$ pads were used to anesthetize the flies and then aligned facing one eye upwards. Using combination of white and blue light (for proper visualization of pseudopupils) from two different light sources under a stereoscopic dissection microscope, each fly was scored for presence or loss of the deep pseudopupil[22], and the percentage of flies with intact pseudopupils was calculated. For each genotype/treatment, over 50 flies per replica and three independent replica were scored blindly and independently by two persons.

## Generation of stable cell lines for purification of Aln pseudokinase and Dipk1C

HEK293A cells stably expressing Aln and Dipk1C were generated as follows. Coding sequences for Aln[34-C] and Dipk1C[51-C], both deleted for the predicted N-terminal transmembrane domains, were cloned into the retroviral pQCXIP vector thereby adding an N-terminal IL-2 signal peptide and C-terminal 3x-Flag tag. 5 μg of the kinase expression vectors and 5 μg of the pCL10A1 helper vector were transfected into HEK293A cells at ~40% confluency. Cells were allowed to divide to >

80% confluency before supplementing the media with 5 μg/mL puromycin. Surviving cells were selected for 3 passages with puromycin before use.

## Immunoprecipitation of constructs

HEK293A cells stably expressing Flag-tagged Aln and Dipk1C were grown to > 80% confluency in 15 cm dishes containing DMEM media supplemented with 10% FBS and Pen/Strep. Cells were washed with 1x PBS once, followed by one wash with serum-free DMEM. Cells were then incubated at 37°C for two days in serum-free media. Subsequently, supernatants were collected and spun at 800xg for 5 min at room temperature. Cleared supernatants were incubated with 50 μL of packed Flag M2 affinity gel (Millipore Sigma A2220) and nutated at 4°C for 1 hr. Beads were spun down at 800xg for 5 min at 4°C and washed once with ice cold 1x TBS and eluted in 50 μL of 100 μM 3x-Flag peptide diluted in 1x TBS.

## in vitro kinase assay

For kinase assays as shown in Fig. 2e, 11 μL of each immunoprecipitated kinase was incubated in 20μL of a buffer containing 50 mM Tris-HCl pH 7.5, 5 mM $MgCl_2$, 100 μM ATP (1000 cpm/pmol [γP$^{32}$]-ATP), and 2.5 mM $Ca^{2+}$. Reactions were incubated at 37°C for 30 min, stopped with 2 μL of 100 mM EDTA and 6 μL of 5x SDS LD with 5% BME. Samples were boiled for 5 min and 2 μL was subjected to SDS-PAGE and stained with Coomassie and incorporation of $^{32}$P was monitored by autoradiography.

## Calcium activation of in vitro kinase activity

Kinase assays as shown in Fig. 2c,d were conducted in 20 μL of a buffer containing 50 mM Tris-HCl pH 7.5, 5 mM $MgCl_2$, 100 μM ATP (1000 cpm/pmol [γP$^{32}$]-ATP), 10 μg Myelin basic protein (MBP), and 3 μg of human Dipk1C kinase domains fused to maltose binding protein (generous gift from the Dixon lab). Calcium concentrations were varied from 0.1 mM to 5 mM. Reactions were incubated at 37°C for 30 min, stopped with 2 μL of 100 mM EDTA and 6 μL of 5x SDS loading dye with 5% β-mercaptoethanol. Samples were boiled for 5 min and 20 μL was subjected to SDS-PAGE and stained with Coomassie. $^{32}$P incorporation was monitored by autoradiography.

## Immunohistochemistry

Fly heads were dissected, and after proboscis removal, immediately transferred to ice-cold 3% glyoxal solution pH 4.0[70] for one hour and subsequently washed overnight in 25% (wt/vol) sucrose in phosphate buffer (pH 7.4), embedded in Optimal Cutting Temperature compound (EMS, Hatfield, PA) frozen in liquid nitrogen and sectioned at 20-μm thickness on a cryostat microtome (CM 1950, Leica Microsystems, Wetzlar, Germany). Sections were washed (PBS with 0.1% triton X100, PBT), blocked (10% NGS) and probed with primary antibody diluted in 5% NGS solution. Primary and secondary antibody dilutions are listed in key resource table. Images were captured with a 20x NA-0.8 or an oil-immersion 63× NA-1.4 lens on an inverted confocal microscope (LSM710 or LSM880 with Airyscan, Carl Zeiss). For each genotype and light condition, immunohistochemistry experiments were performed in three biological replicas with independent sets of flies, using identical acquisition settings. Experimenters were blinded to sample identity before acquiring images or before quantification as appropriate. For Ty1 staining, after primary antibodies (1:500) and 3 washes in PBT, samples were incubated with STAR red secondary antibodies (1:500), washed 3 times and mounted in Abberior mount liquid anti-fade ($n$ = 1.38, pH= 8.5).

## Quantification of fluorescence staining

Fluorescence images were quantified using ImageJ (NIH) adapting previous methods[71] and Imaris 9.5 (Oxford Instruments) software was used for punctate quantification and creating masks. For each antibody, a threshold was determined, removing the lowest 10% of signal

in LD control samples (to reduce variation from low level background signals). This same threshold was applied, and a mask was created for every image in a batch of staining. Before quantification images were blinded for their identity from the user. For the quantification of BiP, ATF4 and Xbp1 in retinas, within 1-μm optical slices, regions were selected manually and assigned as Regions of Interest. The integrated pixel intensity per unit area was measured within this selected area. For each genotype and treatment, more than four flies were quantified from two independent biological replicas randomly mixed during blinding. Data was normalized to the wild-type LD control for each replica. Imaris software was used to quantify ATG8 and Aln-Ty1 punctae in respective images in 3D.

### Western blot

For immunoblotting, 3-5 days aged 10 adult fly heads were homogenized in 100 μl lysis buffer (10% SDS, 6 M urea, and 50 mM Tris-HCl, pH 6.8) and then sonicated for 5 min at high intensity followed by boiling at 95 °C, for 2 min. Lysates were spun for 10 min at 15,000x$g$, supernatant was separated. 5 μl (1/2 heads) of lysates were separated on SDS-PAGE, transferred to nitrocellulose membrane, blocked with 3% non-fat dried milk dissolved in wash buffer (20 mM Tris-Hcl pH 7.5, 5.5 mM NaCl and 0.1 % Tween-20) and probed in primary antibodies in following dilutions; rabbit anti-ATG8a (1:1000), Mouse anti-Ty1 clone BB2 (1:2000), rabbit anti-Hook (1:5000), mouse anti-Actin (JLA20) (1:2000), rabbit anti-RFP (1:1000), Mouse anti-GFP clone B2 (1: 250), and incubated overnight at 4 °C. blots were washed 3 times 10 mins each with wash buffer. Bound antibodies were detected using IR-dye labeled secondary antibodies (1:15,000) and the Odyssey scanner (LI-COR Biosciences) and quantified using EmpiriaStudio V. 2.3.0.154. Pre-stained molecular weight markers (HX Stable) were obtained from UBP-Bio.

### Feeding CQ for autophagy flux

For autophagy flux assays, fly food was prepared by adding chloroquine (CQ) with 1 mg/ml of CQ as final concentration. Flies aged 3 to 5 days were transferred to the CQ-containing food and placed in the LL or LD treatment chambers. After completion of the treatment flies were used for western blot and immunofluorescence.

### Measuring sleep

$CO_2$ anaesthetized, 3-5 days old male flies were placed inside a 65 ×3 mm glass tube (inside-diameter) with standard molasses fly food on one side and a cotton cover on the other side. Tubes were placed in the Ethoscope arena[34,35]. Ethoscopes were placed in a 25 °C incubator with 300 Lux light at a 12 h:12 h LD cycle. After allowing flies to acclimatize for one day, their movement was recorded for 4 days. 5-min periods of inactivity, as determined by video tracking[34,35] were used to define sleep. Finally, Ethoscope data were analyzed using rethomics[34,35]. The data shown in the graphs are averages of the 3-day recordings. All codes used in the sleep analysis are available in github (https://github.com/mz27ethio/Ethoscope-Kramerlab.git).

### Bioinformatics

In order to generate sequence logos for allnighter and canonical kinases, representative sets of kinase-like domains were collected using Allnighter and human protein kinase A, respectively, as BLAST queries against NCBI RefSeq protein sequence database. Sequence sets thus collected were cleared of redundancy at identity threshold 0.8 and aligned using Mafft v.7.463[72]. Protein structures were visualized and analyzed in PyMOL V.2.5.5.

### Statistical analysis

Statistical significance of the results was analyzed using Prism-GraphPad. Anderson-darling or Shapiro-Wilk test was used to assess the normality assumption for all the continuous parameters. Substantially skewed data were transformed on log scale followed by similar normality assessment. Independent t-test was used to compare the normally distributed parameters between two groups whereas the skewed parameter was compared using Mann-Whitney U test.

When the comparative groups were three and more, one-way analysis of variance followed by Bonferroni correction (a multiple comparison test) was used for normally distributed sample parameters. We used two-way analysis of variance followed by Bonferroni's test for multiple comparisons to identify the individual pairwise comparisons to separate the effects of treatment and genetic background. Skewed parameters were compared using Kruskal–Wallis test followed by Dunn's test for multiple comparison test. Bar graphs generated from these analyses demonstrate either means ± SD or median with interquartile range. P values smaller than 0.05 were considered as statistically significant, and values are indicated with one (< 0.05), two (< 0.01), three (< 0.001), or four (< 0.0001) asterisks.

### Reporting summary

Further information on research design is available in the Nature Portfolio Reporting Summary linked to this article.

## Data availability

Sleep data are available in the Zenodo database under accession code (DOI: 10.5281/zenodo.7843827). All other data generated or analyzed during this study are included in this published article (and its supplementary information files). Source data are provided with this paper. Other databases used in this study: NCBI refseq protein; Flybase; Bloomington Drosophila stock center; Source data are provided with this paper.

## Code availability

The modified code to load, clean, and analyze the sleep data was initially adapted from https://rethomics.github.io/[34,35]. Additional code was written to measure Zt-12 latency to sleep. All codes used in this study are deposited at github: https://github.com/mz27ethio/Ethoscope-Kramerlab.git.

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

## Acknowledgements

The authors thank Drs. Dean Smith, Adrian Rothenfluh, Daisuke Hattori and members of the Krämer and Tagliabracci labs for helpful comments to the manuscript and technical assistance. We are grateful for Zuhair Zaidi for the initial setup of ethoscopes, Dr. Quentin Geissmann for advice on ethoscope data analysis, Dr. Byeongha Jeong for his generous help in 3D printing and solving software issues related to ethoscope, and Dr. Shin Yamazaki for kind help in analyzing circadian rhythm data. We thank Dr. Bhaskar Thakur for his kind help in the statistical analysis of data. We thank Dr. Hyung Don Ryoo and the Bloomington Drosophila Stock Center (NIH P40OD018537) for flies, the Molecular and Cellular Imaging Facility at the University of Texas Southwestern Medical Center for help with electron microscopy (NIH S10 OD020103-01). We thank Drs. Hyung Don Ryoo and Joseph M. Bateman, and the Developmental Studies Hybridoma Bank at The University of Iowa for antibodies. This work was funded by NIH grants R01EY010199, R01EY033184 and 5R01AI155426 to H.K. and NIH grant DP2 OD027405 and Welch Foundation Grant I-1911 to V.S.T.

## Author contributions

S.S.: conceived, designed and performed experiments, statistical analysis, and analyzed data A.T.M.: conceived, designed and performed experiments, statistical analysis, and analyzed data B.P.: designed and performed biochemical experiments, M.E.: performed some ERG and sleep experiments and adapted codes for sleep measurements, C.T.: performed qPCR experiments, I.T.: gave input and edited the manuscript. K.P.: performed bioinformatic analysis and conceived experiments V.S.T.: conceived and supervised research and edited the manuscript HK.: conceived, designed experiments, analyzed data, and supervised the research. S.S. and H.K. wrote the manuscript with input from all co-authors

## Competing interests

The authors declare no competing interests.
