## [Peer Review File · Nature Communications]

Allnighter pseudokinase-mediated feedback links dysregulated proteostasis and sleep in *Drosophila*REVIEWER COMMENTS

Reviewer #1 (Remarks to the Author):

This is an interesting manuscript on the roles of the Allnighter (Aln) pseudokinase in the *Drosophila* photoreceptors and brain, showing many parallels with the related Dipk proteins.

This is a dense manuscript with dense and often difficult to follow figures. The results are generally convincing, with some suggestions for clarifications and additional studies below. [SEP]

Specific comments:

Statistical methods aren't detailed.

The figure legends are very terse. This reviewer finds papers easier to read if there is sufficient text in the legends that they can pretty much explain the experiments with minimal reading of the body of the results. This isn't the case here. [SEP]

I'm not finding the gene CG12308=aln in the bible of fly genes, Flybase. Is this CG# correct? or has the CG# changed with a late release of genome data?

Fig 1 nicely shows that rhabdomeric area and several proteins are reduced by 3 days LL. But constant light will also stop the circadian clock in flies. The easiest way to rule out an effect due to altered clock function would be to repeat some of these measurements in a clock defective fly.

Fig 3. Most of the effects in this figure are really small. Is this figure really necessary?
See comments above about possible circadian effects in LL.

Is ANOVA the appropriate stat test here? Some of the data looks very non-normally distributed. I'm having a hard time understanding the time course data. In E, why is there an increase in WT only between LD & day 1 LL? And in aln2, perhaps a decrease between LD and day 3 LL. I'm left wondering if there's anything here except that aln2 are overall a bit higher than WT. This figure is confusing to a non-expert in UPR physiology.

Fig 3O shows the largest effect in this figure but only in heads. Why not measure this in eyes?

Fig 4: Aln is a secreted protein retrogradely taken up by photoreceptors. [SEP] This is cool.

To confirm the functional significance of these results, would expression of aln in lamina rescue the photoreceptor phenotypes of the aln mutant?

Fig 5: For the experiments expressing shibire, it's unclear what temperature is used. The best controls on these experiments would be restrictive vs permissive temperatures for shibire.

This figure is a difficult read- it might be easier if the conclusion were given as the title for the text

paragraph and figure.

Fig 6: The data show alterations both in day and night sleep in LD conditions, but from the data presented, we can't tell whether this effects is from altered circadian function vs altered responses to light. The analysis shown should be continued into constant dark (DD) conditions to resolve this issue.

Reviewer #2 (Remarks to the Author):

Progress in recent decades has revealed many cell-autonomous mechanisms of proteostasis. At the same time, there has been much intrigue about the non-autonomous proteostatic signaling pathways that affect aging and neurodegenerative disorders. However, our understanding of the non-autonomous mechanisms remains rudimentary. Here, the authors report that an uncharacterized secretory protein, Aln, non-autonomously promotes proteostasis in the fly eye. Aln encodes a kinase domain, but the catalytic residues are dispensable for its proteostatic function. Aln promoter is not active in the fly retina, but the protein is detected in those cells at high levels. Such Aln protein accumulation in the photoreceptor requires endocytosis. In the absence of Aln, the fly photoreceptors become more vulnerable to light-induced stress. In turn, photoreceptor activity non-autonomously regulates Aln expression in the brain through a feedback loop. In addition, the authors show that Aln affects sleep in *Drosophila*.

This study is overall very interesting because it identifies a novel secretory molecule with essential functions in proteostasis and sleep. The experiment is well designed. The data are clear, with proper controls. The results nicely quantified. I only have relatively minor questions for the authors.

1. In Figure 5A-C, the authors show that Aln uptake by the photoreceptors is reduced in *Shi^{ts}* background. Does this affect photoreceptor proteostasis? One could imagine two different outcomes: (1) Aln actually signals through a receptor at the plasma membrane, and Aln endocytosis does affect that proteostatic signaling. (2) Aln needs to be endocytosed act somewhere within cells, in which case *Shi^{ts}* would block Aln-mediated proteostasis.

2. The authors write in the text that they examined sleep "because the circadian change in visual input contributes to circadian rhythm." Now, the data included here clearly shows that Aln affects sleep. But it does not show whether Aln affects sleep through the visual system. I wonder if the authors could experimentally test this (e.g. by making eye specific Aln loss). Alternatively, the authors should discuss the idea that Aln could acts in cells outside the visual system to affect sleep.

Reviewer #3 (Remarks to the Author):

The authors identified Aln pseudokinase that is retrogradely secreted from the second-order neuron of *Drosophila* photoreceptor neurons and endocytosed by the photoreceptors. There, Aln is required to

maintain the activity dependent structural dysmorphology of rhabdomeres. Endogenously Aln is upregulated in optic lobe cells upon prolonged ambient light exposure, and functions in maintaining proteostasis in both the optic lobe neurons and photoreceptors. The authors also tried behavioural analysis and found that the lack of Aln shortens the night sleep, which indicates the link between dysregulated proteolysis and sleep, two features of ageing and neurological diseases. The manuscript is well written and the presented data are of high quality. The generation of novel sophisticated genetic tools of Aln allows a detailed analysis of the localization together with the molecular function and provides a new level of understanding about the molecular link between proteostasis and the regulation of sleep. However, regarding the proposed mechanism, the manuscript could be more convincing by addressing some issues regarding the non-autonomous retrograde transmission of Aln protein and importance of activity dependent regulation of Aln protein level in lamina neurons.

major criticism:1) Although I found the retrograde secretion/endocytosis mechanism is both interesting and important in this study, considering that the author has underlined in the abstract that they found "Aln as a cell non-autonomous regulator of proteostasis responses necessary for normal sleep and structural plasticity of Drosophila photoreceptors.", I also found that the data indicating that Aln is retrogradely transported from the optic lobe neuron to the photoreceptor neuron is rather weak. The authors use Aln-T2A-Gal4 to monitor the expression both in the retina and the lamina neurons, they use membrane localization marker mCD8GFP to see the co-localization with nuclear protein Elav. They should have used NLS-GFP to see the overlap more precisely. I would also suggest using GMR-Gal80 to completely suppress the expression of Aln-GAL4 in the retina to see whether the retinal signal of the GFP signal is non-cell autonomous. To convincingly show the retrograde transport, usually the overexpression of tagged Aln in specific cell populations is done. I would suggest to specifically express UAS-Aln-FLAG only in lamina neurons (or optic lobe neurons) and see whether FLAG signal can be seen in the retina.2) While the author emphasize that activity dependent feedback regulation is critical for the Aln function, the data provided fails to demonstrate that this feedback is biologically and functionally crucial. To see whether the activity dependent upregulation of Aln in lamina neuron is required for preventing dysmorphology of the rhabdomeres, I would suggest the authors to check on ort mutant flies, where Histamine receptor is missing in the second-order neurons in the optic lobe. Alternatively, CarT>TTL flies are also fine. There should be a downregulation of retinal Aln level, and failure in rescuing the dysmorphology of the rhabdomeres/ pseudopupils.

minor points:1) Is the Aln expression in lamina upregulated in LL? If so, does it affect the protein level of Aln in the retina?(Although blocking the activity of photoreceptor cells is shown in Fig.5, activity dependent upregulation of Aln in retinal cells and/or lamina neurons is not shown (in LL).)2) UPR and related jargons must be explained somewhere.3) typo: Fig4. top part (above Band B') UAS-mCD-GFP must be UAS-mCD8-GFP4)Fig3 D" seems to be upregulated although the quantification sdoes not show that in Fig3E. Changing the image to a more representative one could be easier for the readers to understand.5) Please indicate the temperature itself and the timing of the temperature shift for CarT-Shi[ts] experiments. Or is this done in constant 25 degree? Then the temperature could be higher to have a more solid effect.6) Is the retinal signal of Ty1 in Fig5J or L reduced compared to Fig5H? If the feedback loop is critical, it should affect the retinal take up of Aln. Please quantify.7) in the Fig2F, the

Aln[Ty1-T2A-GAl4] should be indicated in the panel in purple letters, to be consistent with other constructs.

Response to reviewer comments

We appreciate to positive comments by the reviewers and the many helpful suggestions and concerns raised. We believe these comments have helped us to greatly improve the manuscript. Especially experiments that show (i) the cell-type specific secretion of Aln from L4 lamina neurons and (ii) the functional rescue by lamina-specific, but not photoreceptor-specific expression significantly enhanced the manuscript.

Reviewer #1 (Remarks to the Author):

This is an interesting manuscript on the roles of the Allnighter (Aln) pseudokinase in the *Drosophila* photoreceptors and brain, showing many parallels with the related Dipk proteins. This is a dense manuscript with dense and often difficult to follow figures. The results are generally convincing, with some suggestions for clarifications and additional studies below. SEP

Specific comments:

1. Statistical methods aren't detailed.

We now have added a specific section describing statistical methods to the Method section and also detailed specific tests used in the figure legends.

2. The figure legends are very terse. This reviewer finds papers easier to read if there is sufficient text in the legends that they can pretty much explain the experiments with minimal reading of the body of the results. This isn't the case here. SEP

We have re-written the figure legends to highlight the major findings of the figures.

3. I'm not finding the gene CG12308=aln in the bible of fly genes, Flybase. Is this CG# correct? or has the CG# changed with a late release of genome data?

We apologize and are grateful to the reviewers for catching this misleading typo: it is actually CG12038., This is now corrected in the paper.

4. Fig 1 nicely shows that rhabdomic area and several proteins are reduced by 3 days LL. But constant light will also stop the circadian clock in flies. The easiest way to rule out an effect due to altered clock function would be to repeat some of these measurements in a clock defective fly.

We appreciate this interesting comment. In response we analyze effects of *per*⁰¹ null flies in the result section: "*Induction of the aln phenotype by the switch from LD to LL strongly suggests an involvement of the circadian system, in line with the disruption of the circadian clock observed under sustained high light intensities*²⁵. Therefore, we wondered whether the Aln LL phenotypes may be mimicked in the *per*⁰¹ circadian clock mutant. However, with regard to Rh1 levels and autophagosome induction, *per*⁰¹ mutant under LD were indistinguishable from wild type or *aln*² under LD, with similar reduction in Rh1 and elevation of ATG8 punctae under LL (Supplementary Fig. 7)."

5. Fig 3. Most of the effects in this figure are really small. Is this figure really necessary? See comments above about possible circadian effects in LL.

We have simplified Fig. 3 by focusing on the first day of LL. At that point, wild type shows significant changes in ER stress responses, which we know from previous work to be critical for normal structural adaptation of photoreceptors (PMID: 30015618). ER stress in *aln* mutants is already high under LD, however, and not further elevated upon switch to LL. The elevated ER stress in *aln* mutants even under LD, assures that this cannot simply be a consequence of possible circadian effects in LL as the circadian periodicity of *aln²* mutants is unaltered compared to wild-type controls (see Supplementary Fig. 10).

6. Is ANOVA the appropriate stat test here? Some of the data looks very non-normally distributed.

We agree and have reanalyzed the data using non-parametric tests. Specific test used are mentioned in the figure legends.

I'm having a hard time understanding the time course data. In E, why is there an increase in WT only between LD & day 1 LL? And in *aln2*, perhaps a decrease between LD and day 3 LL. I'm left wondering if there's anything here except that *aln2* are overall a bit higher than WT. This figure is confusing to a non-expert in UPR physiology.

As mentioned above we now focus only on the physiological important rise from LD to day 1 of LL. On subsequent days, different elements of the UPR (BiP, XBP1-GFP and ATF-5"UTR-DsRed) seem to adapt in slightly different patterns that we cannot explain at this time.

Fig 3O shows the largest effect in this figure but only in heads. Why not measure this in eyes?

We agree. In addition to the quantification in whole heads (now Figure 3P) we also added the quantification in retina (now Figure 3O).

Fig 4: Aln is a secreted protein retrogradely taken up by photoreceptors. ^[1]_[Sep] This is cool. To confirm the functional significance of these results, would expression of *aln* in lamina rescue the photoreceptor phenotypes of the *aln* mutant?

We appreciate the suggestion and have now included the results of this experiment using the lamina-specific *ort-Gal4* driver in the new Figure 6. Indeed lamina-specific expression rescued the *aln²* phenotype (new Figure 6 I), in contrast to photoreceptor-specific expression. A further, related surprise is the cell-type specific secretion of Aln from L4 neurons, but not L2 (documented in the new Fig. 6A-D).

Fig 5: For the experiments expressing *shibire*, it's unclear what temperature is used. The best controls on these experiments would be restrictive vs permissive temperatures for *shibire*.

We now specify that the experiments were performed at 25°C. We agree that typically *Shi^[ts1]* is used at higher temperatures to completely block endocytosis. However, it is important to note that inhibitory effects in response to photoreceptor-

specific expression of Shi[ts1] have been observed at temperatures as low as 19°C (PMID: 19906968). Furthermore, we have some preliminary indications that Allnighter expression may be affected by temperature and cellular stress responses. As the experiments in Figure 5 are specifically aimed at evaluating long-term changes in Aln expression, we preferred to perform these experiments at constant temperature. Importantly, an independent method - photoreceptor-specific expression of Tetanus toxin light chain (TTL) confirmed the key finding of altered expression of Aln in L4 lamina neurons upon interfering with synaptic activity of photoreceptors.

This figure is a difficult read- it might be easier if the conclusion were given as the title for the text paragraph and figure.

We have rewritten all figure legend, to include the key messages of each figure in its legend.

Fig 6: The data show alterations both in day and night sleep in LD conditions, but from the data presented, we can't tell whether this effect is from altered circadian function vs altered responses to light. The analysis shown should be continued into constant dark (DD) conditions to resolve this issue.

Supplementary Fig. 10 DD in the revised manuscript shows an actogram of wild type and *aln²* flies in LD and DD conditions. We find the circadian periodicity of *aln²* mutants is unaltered compared to wild-type controls. Furthermore, our ERG recordings under LD conditions do not reveal altered responses to light in *aln²* mutants. We thus favor the notion that *aln* has a function in the regulation of sleep. Identifying specific mechanisms and cell types responsible for the phenotypes would further address a possible role of the circadian clock. However, those experiments are beyond the scope of this initial analysis of *aln* function.

Reviewer #2 (Remarks to the Author):

Progress in recent decades has revealed many cell-autonomous mechanisms of proteostasis. At the same time, there has been much intrigue about the non-autonomous proteostatic signaling pathways that affect aging and neurodegenerative disorders. However, our understanding of the non-autonomous mechanisms remains rudimentary. Here, the authors report that an uncharacterized secretory protein, Aln, non-autonomously promotes proteostasis in the fly eye. Aln encodes a kinase domain, but the catalytic residues are dispensable for its proteostatic function. Aln promoter is not active in the fly retina, but the protein is detected in those cells at high levels. Such Aln protein accumulation in the photoreceptor requires endocytosis. In the absence of Aln, the fly photoreceptors become more vulnerable to light-induced stress. In turn, photoreceptor activity non-autonomously regulates Aln expression in the brain through a feedback loop. In addition, the authors show that Aln affects sleep in *Drosophila*.

This study is overall very interesting because it identifies a novel secretory molecule with essential functions in proteostasis and sleep. The experiment is well designed. The data are clear, with proper controls. The results nicely quantified. I only have relatively minor questions for the authors.

1. In Figure 5A-C, the authors show that Aln uptake by the photoreceptors is reduced in Shi^{ts} background. Does this affect photoreceptor proteostasis? One could imagine two different outcomes: (1) Aln actually signals through a receptor at the plasma membrane, and Aln endocytosis does affect that proteostatic signaling. (2) Aln needs to be endocytosed act somewhere within cells, in which case Shi^{ts} would block Aln-mediated proteostasis.

We appreciate this interesting question, and have tested this possibility by staining for ATF4, a target of the UPR. Indeed, as shown in Supplementary Fig. 9H,I and quantified in panel Supplementary Fig 9J. We observed elevated levels of ATF4 in the retinas of flies expressing Shi^[ts1] consistent with increased proteostasis stress. This result, seeming to support the first possibility mentioned by the reviewer, was also consistent with the new data shown in Figure 6 that lamina-specific, but not photoreceptor-specific expression of uas-Aln-Flag is sufficient to rescue the *aln*² visual phenotype under LL.

2. The authors write in the text that they examined sleep “because the circadian change in visual input contributes to circadian rhythm.” Now, the data included here clearly shows that Aln affects sleep.

But it does not show whether Aln affects sleep through the visual system. I wonder if the authors could experimentally test this (e.g. by making eye specific Aln loss).

Alternatively, the authors should discuss the idea that Aln could acts in cells outside the visual system to affect sleep.

In preliminary experiments we detected Aln expression in sleep-related brain areas in addition to the mushroom body which can be seen in Figure 4A. However, as we have not yet determined the specific cell types expressing Aln in those areas, and consider this beyond the scope of the current manuscript, we felt these data were too preliminary to include in the current manuscript. We now address this issue in the discussion by stating: “*Furthermore, it remains to be determined whether the effect of Aln on sleep loss reflects its function in the visual system or its expression in other brain regions, including the mushroom body as seen in Figure 4 .*”

Reviewer #3 (Remarks to the Author):

The authors identified Aln pseudokinase that is retrogradely secreted from the second-order neuron of *Drosophila* photoreceptor neurons and endocytosed by the photoreceptors. There, Aln is required to maintain the activity dependent structural dysmorphology of rhabdomeres. Endogenously Aln is upregulated in optic lobe cells upon prolonged ambient light exposure, and functions in maintaining proteostasis in both the optic lobe neurons and photoreceptors. The authors also tried behavioral analysis and found that the lack of *aln* shortens the night sleep, which indicates the link between dysregulated proteolysis and sleep, two features of ageing and neurological diseases.

The manuscript is well written and the presented data are of high quality. The

generation of novel sophisticated genetic tools of Aln allows a detailed analysis of the localization together with the molecular function and provides a new level of understanding about the molecular link between proteostasis and the regulation of sleep. However, regarding the proposed mechanism, the manuscript could be more convincing by addressing some issues regarding the non-autonomous retrograde transmission of Aln protein and importance of activity dependent regulation of Aln protein level in lamina neurons.

major criticism:

1) Although I found the retrograde secretion/endocytosis mechanism is both interesting and important in this study, considering that the author has underlined in the abstract that they found "Aln as a cell non-autonomous regulator of proteostasis responses necessary for normal sleep and structural plasticity of *Drosophila* photoreceptors.", I also found that the data indicating that Aln is retrogradely transported from the optic lobe neuron to the photoreceptor neuron is rather weak.

The authors use Aln-T2A-Gal4 to monitor the expression both in the retina and the lamina neurons, they use membrane localization marker mCD8GFP to see the co-localization with nuclear protein Elav. They should have used NLS-GFP to see the overlap more precisely.

This experiment is now included in Supplementary Figure 8C and adds further confirmation to the neural nature of the Aln expressing cells. We would like to point out, however, that the benefit of using mCD8GFP is outlining the entire cell, which is especially advantageous in distinguishing the Aln-positive bristle neurons in the retina from the photoreceptor neurons with their close-by nuclei as seen in Figure 4A,B G,H.

I would also suggest using GMR-Gal80 to completely suppress the expression of Aln-GAI4 in the retina to see whether the retinal signal of the GFP signal is non-cell autonomous.

We now expressed NLS-mCherry under control of Aln-Gal4 with and without ey-Gal80 and did not see a significant difference other than loss of expression in bristle neurons and their axons, as highlighted in the high exposure images in Figure 4G',H'. We feel a stronger case for secretion of Aln from lamina neurons and for the necessity of this secretion, as opposed to some low-level retinal expression, comes from new experiments (now shown in the new Figure 6) which are based in this reviewer's excellent suggestion of lamina-specific expression (see below).

To convincingly show the retrograde transport, usually the overexpression of tagged Aln in specific cell populations is done. I would suggest to specifically express UAS-Aln-FLAG only in lamina neurons (or optic lobe neurons) and see whether FLAG signal can be seen in the retina.

We are grateful for this excellent suggestion. This experiment, now shown in the new Figure 6, allowed us to conclude that secretion from the lamina to the retina occurs,

but is cell-type specific as only L4, but not L2 specific expression yield retinal uptake of Aln-Flag.

2) While the author emphasize that activity dependent feedback regulation is critical for the Aln function, the data provided fails to demonstrate that this feedback is biologically and functionally crucial.

To see whether the activity dependent upregulation of Aln in lamina neuron is required for preventing dysmorphology of the rhabdomeres, I would suggest the authors to check on ort mutant flies, where Histamine receptor is missing in the second-order neurons in the optic lobe. Alternatively, CarT>TTL flies are also fine. There should be a downregulation of retinal Aln level and failure in rescuing the dysmorphology of the rhabdomeres/ pseudopupils.

Based on these suggestions, we added several experiments to further strengthen the conclusion that Aln participates in a feedback loop that is biologically and functionally crucial. First, as suggested we show that CarT>TTL, as well as the CarT[43] null mutant (which we previously showed to be defective in histamine recycling; PMID: 26653853) interfere with ommatidial structural integrity as assessed by DPP. This is shown in the new Figure 5O. Second, we show that levels of Aln expression, in addition to CarT>Shit[ts1] and CarT>TTL, are also sensitive to interference with the standard LD condition, by placing flies into LL or DD (new Figure 5P-T). Third, we show that lamina-specific expression of Aln-Flag, but not photoreceptor specific expression, is sufficient to rescue the vision defect of *aln*² (new Figure 6E-I). Together our data show that Aln expression in the lamina - directly or indirectly - is responsive to changes in visual activity and is necessary for the maintenance of structural plasticity in the visual system.

minor points:

1) Is the Aln expression in lamina upregulated in LL? If so, does it affect the protein level of Aln in the retina? (Although blocking the activity of photoreceptor cells is shown in Fig.5, activity dependent upregulation of Aln in retinal cells and/or lamina neurons is not shown (in LL).

Surprisingly, we find that both DD and LL cause downregulation of Aln expression and staining for Aln in the retina (shown in new Figure 5 P-T).

2) UPR and related jargons must be explained somewhere

We now more explicitly introduce key details necessary for the in-vivo UPR analysis in the Result section: "*The UPR is triggered by accumulation of misfolded proteins in the ER. In response, activated Ire1 promotes splicing of the transcription factor XBP1 as can be measured with a XBP1-GFP fusion 24 and activated PERK kinase phosphorylates eIF2 thereby reducing general translation but upregulating ATF4 translation as measured by a ATF4-5'UTR-DsRed fusion 25. The combined activity of the activated XBP1, ATF4 and ATF6 transcription factors promotes the expression of a multitude of factors supporting protein folding that aid in restoring homeostasis. We previously showed that homeostatic UPR is a protective process necessary for photoreceptor adaptation (5) "*

.3) typo: Fig4. top part (above Band B') UAS-mCD-GFP must be UAS-mCD8-GFP
Thank you for catching this. We have changed this in the revised version of manuscript.

4) Fig3 D" seems to be upregulated although the quantification does not show that in Fig3E. Changing the image to a more representative one could be easier for the readers to understand.

We re-focused the UPR data in the revised Figure 3 on the critical period of LD to 1 day of LL when the major changes occur. Thus the Fig.3D" is no longer included.

5) Please indicate the temperature itself and the timing of the temperature shift for CarT-Shi[ts] experiments. Or is this done in constant 25 degree? Then the temperature could be higher to have a more solid effect.

We now explicitly state that the experiment was conducted at 25°C "*Experiments employing CarT[HA-T2A-QF2]-driven QUAS-Shi[ts1] expression to inhibit endocytosis were conducted at 25°C, a temperature effective in photoreceptors as confirmed by ERGs and consistent with previous results showing that Shits1 expression can change photoreceptor morphology as low as 19°C (ref. 65)*". The reasons are discussed above. Note that the effect of CarT> Shi[ts1] was as strong as that of CarT>TTL (see Figure 5 G-N).

6) Is the retinal signal of Ty1 in Fig5J or L reduced compared to Fig5H? If the feedback loop is critical, it should affect the retinal take up of Aln. Please quantify.

Retinal Aln-Ty1 punctae are now quantified in Figure 5N and they are indeed reduced.

7) in the Fig2F, the Aln[Ty1-T2A-GAI4] should be indicated in the panel in purple letters, to be consistent with other constructs.

The name of the Aln[Ty1-T2A-GAI4] allele has been added to Figure 2F.

REVIEWERS' COMMENTS

Reviewer #1 (Remarks to the Author):

The revised version nicely addresses all the issues brought up at review, resulting in a much improved manuscript.

Two small points:

Fig 3O: 'Retian'=> Retina

193 Induction of the aln phenotype by the switch from LD to LL strongly suggests an
194 involvement of the circadian system, in line with the disruption of the circadian clock observed
195 under sustained high light intensities 27.

'strongly' seems a bit strong here since this isn't what is found.

Reviewer #2 (Remarks to the Author):

The authors have identified a novel non-autonomous regulator of proteostasis in *Drosophila*. They have satisfactorily addressed the points that I had raised. Therefore, I support the publication of this manuscript.

Reviewer #3 (Remarks to the Author):

The authors have now answered to all the points I raised, and the manuscript is now highly convincing. I support publication of the manuscript in *Nature Communications*.

There is one typo I noticed in the figure panel.

Figure 3O, "Retian" should be "Retina".

Point-by-point response to the REVIEWERS' COMMENTS

We appreciate the reviewers' comments and have addressed the issues raised.

Reviewer #1 (Remarks to the Author):

The revised version nicely addresses all the issues brought up at review, resulting in a much improved manuscript.

Two small points:

Fig 3O: 'Retian'=> Retina

We corrected this typo.

193 Induction of the aln phenotype by the switch from LD to LL strongly suggests an
□194 involvement of the circadian system, in line with the disruption of the circadian
clock observed

□195 under sustained high light intensities 27. □

'strongly' seems a bit strong here since this isn't what is found.

We agree and removed "strongly"

Reviewer #2 (Remarks to the Author):

The authors have identified a novel non-autonomous regulator of proteostasis in *Drosophila*. They have satisfactorily addressed the points that I had raised. Therefore, I support the publication of this manuscript.

Reviewer #3 (Remarks to the Author):

The authors have now answered to all the points I raised, and the manuscript is now highly convincing. I support publication of the manuscript in *Nature Communications*.

There is one typo I noticed in the figure panel.

Figure 3O, "Retian" should be "Retina".

We corrected this typo.